# Thousands of novel translated open reading frames in humans inferred by ribosome footprint profiling

**Anil Raj[1]\*[†], Sidney H Wang[2]\*[†], Heejung Shim[2†‡], Arbel Harpak[3], Yang I Li[1], Brett Engelmann[2], Matthew Stephens[2,4], Yoav Gilad[2]\*, Jonathan K Pritchard[1,3,5]\***

[1]Department of Genetics, Stanford University, Stanford, United States; [2]Department of Human Genetics, University of Chicago, Chicago, United States; [3]Department of Biology, Stanford University, Stanford, United States; [4]Department of Statistics, University of Chicago, Chicago, United States; [5]Howard Hughes Medical Institute, Stanford University, Stanford, United States

**Abstract** Accurate annotation of protein coding regions is essential for understanding how genetic information is translated into function. We describe riboHMM, a new method that uses ribosome footprint data to accurately infer translated sequences. Applying riboHMM to human lymphoblastoid cell lines, we identified 7273 novel coding sequences, including 2442 translated upstream open reading frames. We observed an enrichment of footprints at inferred initiation sites after drug-induced arrest of translation initiation, validating many of the novel coding sequences. The novel proteins exhibit significant selective constraint in the inferred reading frames, suggesting that many are functional. Moreover, ~40% of bicistronic transcripts showed negative correlation in the translation levels of their two coding sequences, suggesting a potential regulatory role for these novel regions. Despite known limitations of mass spectrometry to detect protein expressed at low level, we estimated a 14% validation rate. Our work significantly expands the set of known coding regions in humans.

**\*For correspondence:** rajanil@ stanford.edu (AR); siddisis@ uchicago.edu (SHW); gilad@ uchicago.edu (YG); pritch@ stanford.edu (JKP)

[†]These authors contributed equally to this work

**Present address:** [‡]Department of Statistics, Purdue University, West Lafayette, United States

**Competing interests:** The authors declare that no competing interests exist.

## Introduction

Annotations for coding sequences (CDSs) are fundamental to genomic research. The GENCODE Consortium (*Harrow et al., 2012*) has played an important role in annotating the set of protein coding sequences in the human genome, predominantly relying on manual annotation from the Human and Vertebrate Analysis and Annotation (HAVANA) group (*Wilming et al., 2008*). Their annotation pipeline identifies coding sequences using homology with sequences in large cDNA/EST databases and the SWISS-PROT protein sequence database (*Bairoch and Apweiler, 2000*), and validates them using sequence homology across vertebrates and using tandem mass spectrometry. Despite being the most comprehensive database of CDSs available, the current set is conservative and does not include several classes of CDSs, including translated upstream open reading frames (ORFs), dually coded transcripts, and N-terminal extensions and truncations.

Recent work has made it increasingly clear that much of the human genome is transcribed in at least one tissue during some stage of development (*Hangauer et al., 2013*; *Djebali et al., 2012*; *Birney et al., 2007*; *Clark et al., 2011*; *Kapranov et al., 2007*; *van Bakel et al., 2010*). However, the functional implication for most of these transcripts remains unclear; in particular, the set of sequences translated from these transcripts are not yet completely characterized. For example, there are several recent studies in which RNA transcripts that were previously annotated as noncoding were shown to encode short functional peptides. One well characterized example is the *polished*

rice (pri) / tarsal-less (tal) locus in flies, a polycistronic mRNA encoding four short peptides active during embryogenesis (*Kondo et al., 2007*, *2010*; *Galindo et al., 2007*). While short peptides are known to play critical roles in multiple biological processes (*Lauressergues et al., 2015*; *Oelkers et al., 2008*; *Le Mercier et al., 2006*; *Jung et al., 2009*), annotating genomic regions that encode them remains challenging.

Direct proteogenomic mass spectrometry has the potential to fill this gap but suffers from variable accuracy in assignment of peptide sequences to spectra and assignment of identified peptides to proteins (for peptides shared across database entries). Moreover, these approaches suffer from a "needle in a haystack" problem when searching all six translational frames over the transcribed portion of the genome (*Nesvizhskii, 2014*; *Le Mercier et al., 2006*; *Ma, 2015*). Alternative approaches that utilize empirically-derived phylogenetic codon models to distinguish coding transcripts from non-coding transcripts are promising (*Lin and Kellis, 2011*). However, the success of such approaches is contingent on the duration, strength and stability of purifying selection and these methods may be underpowered for short coding sequences or for newly evolved coding sequences.

Ribosome profiling utilizes high throughput sequencing of ribosome-protected RNA fragments (RPFs) to quantify levels of translation (*Ingolia et al., 2009*). Briefly, the technique consists of isolating monosomes from RNase-digested cell lysates and extracting and sequencing short mRNA fragments protected by ribosomes. Early studies of ribosome profiling have shown that RPFs are substantially more abundant within the CDS of annotated transcripts compared to the 5' or 3' untranslated regions (UTRs) (*Ingolia et al., 2009*; *Weinberg et al., 2016*). More importantly, when aggregated across annotated coding transcripts, the RPF abundance within the CDS has a clear three base-pair periodicity while the RPF abundance in the UTRs lacks this periodic pattern.

Recently, using ribosome profiling data, several studies reported conflicting results on the coding potential of long intergenic noncoding RNA (*Ingolia et al., 2011*; *Guttman et al., 2013*; *Ingolia et al., 2014*). These studies assessed coding potential using either i) the enrichment of RPFs within the translated CDS relative to background, or ii) the difference in length of RPFs within the translated CDS compared to background. However, these approaches may lack power for several reasons. First, they make little distinction between ribosomes scanning through the transcript and ribosomes decoding the message. Second, the enrichment signal can be severely diminished if the transcript is significantly longer than the coding region within it. Third, there is often substantial variance in RPF abundance within the CDSs, which can decrease power to detect translated sequences when using a simple RPF enrichment statistic alone. Other studies have used the periodicity structure in RPF counts to identify dual coding sequences and short translated CDSs (*Michel et al., 2012*; *Bazzini et al., 2014*), but the methods reported high false positive rates and could only identify a few hundred CDSs.

In this work, we developed riboHMM; a model to identify translated CDSs by leveraging both the total abundance and the codon periodicity structure in RPFs. We used this model to identify thousands of novel CDSs in the transcriptome of human lymphoblastoid cell lines (LCLs).

## Probabilistic model to infer translated coding sequences

Ribosome footprint profiling data, when aggregated across annotated coding transcripts centered at their translation initiation (or termination) sites (*Figure 1A*), show two distinct features that distinguish the CDS from untranslated regions (UTRs).

- **Higher abundance within the CDS**. RPF counts are highly enriched within the CDS overall. Moreover, base positions within the CDS close to the translation initiation and termination sites have substantially higher RPF counts compared to base positions in the rest of the CDS. Untranslated regions have very low RPF counts, with the 5'UTR having a slightly higher RPF count compared to the 3'UTR. Furthermore, base positions in the 5'UTR immediately flanking the initiation site have a slightly higher RPF count compared to the rest of the 5'UTR; a similar pattern is observed in the 3'UTR.
- **Three-base periodicity within the CDS**. RPF counts typically peak at the first position of each codon. The RPF count over the initiation and termination codons tend to have a stronger peak (thus, a slightly different periodic pattern) compared to the rest of the CDS. The RPF counts in the UTRs lack this periodic pattern with similar aggregate counts among base positions in the 5'UTR and 3'UTR.

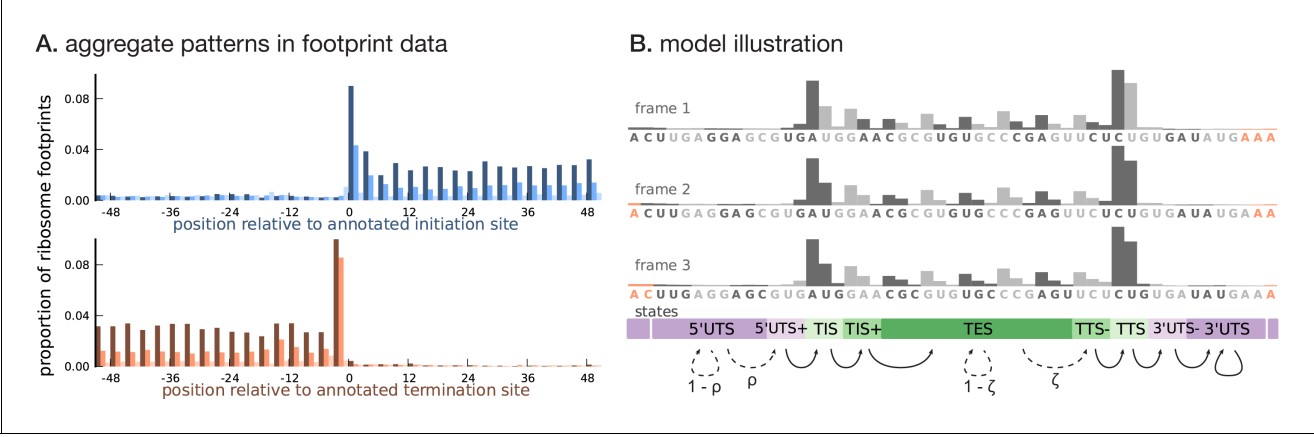

**Figure 1.** Illustrating the model. (**A**) Proportion of footprint counts aggregated across 1000 highly expressed annotated coding transcripts, centered at their translation initiation (blue) and termination (orange) sites. In aggregate, RPF count data have higher abundance within the CDS than the UTRs and exhibit a 3-base periodicity within the CDS. (**B**) Each transcript belongs to one of three unobserved reading frames, and is represented as a sequence of base-triplets (highlighted by differing shades of gray) that depends on the reading frame. Each triplet belongs to one of nine unobserved states. The state sequence shown corresponds to frame 3 and varying shades from purple to green highlight the different states. Base positions marked in orange are modeled independently and always belong to the relevant UTS state. Transitions with nonzero probabilities are indicated by arrows, with solid arrows denoting a probability of 1 and dotted arrows denoting probabilities that are a function of the underlying sequence.

The following figure supplements are available for figure 1:

**Figure supplement 1.** Robustness of periodicity parameter estimates.

**Figure supplement 2.** Robustness of occupancy parameter estimates.

**Figure supplement 3.** Decision rules to identify matches and mismatches of mCDS to annotation.

**Figure supplement 4.** Model accuracy.

**Figure supplement 5.** Comparing footprint abundance and gene expression.

**Figure supplement 6.** Comparing the periodicity in ribosome footprint counts for footprints of different lengths.

**Figure supplement 7.** Robustness of parameters for start codon usage to choice of learning set.

We developed a framework to infer the translated CDS in a transcript using a model that 1) captures these distinct features of ribosome profiling data and 2) integrates RNA sequence information and transcript expression. As illustrated in *Figure 1B*, to capture the three-base structure in the RPF count data within the CDS, we represented a transcript as a sequence of non-overlapping base triplets. The CDS of the transcript is required to belong to one of three reading frames. To account for all three reading frames, each transcript has a latent frame variable that specifies at which base position of the transcript we begin enumerating the triplets.

Conditional on the frame, we modeled the data for each transcript, represented as a sequence of base triplets, using a hidden Markov model (HMM). Each triplet belongs to one of nine latent states — 5'UTS (5' Untranslated State), 5'UTS+ (the last untranslated triplet prior to the initiation site), TIS (Translation Initiation State), TIS+ (the triplet immediately following the initiation site), TES (Translation Elongation State), TTS- (the translated triplet prior to the termination site), TTS (Translation Termination State), 3'UTS- (the first untranslated triplet immediately following the termination site), and 3'UTS (3' Untranslated State). The states {TIS, TIS+, TES, TTS-, TTS} denote translated triplets and {5'UTS, 5'UTS+, 3'UTS-, 3'UTS} denote untranslated triplets. The probability distribution over the possible sequence of latent states is a function of the underlying RNA sequence. *Figure 1B* illustrates these states, and how they relate to each other, in conjunction with the transcript representation. The groups

of states {5'UTS+, TIS, TIS+} and {TTS-, TTS, 3'UTS-} help model the distinct structure of the RPF counts around the translation initiation and termination sites, respectively.

Assuming each transcript has either 0 or 1 CDS, we restricted the possible transitions between latent states as shown in *Figure 1B*: transitions from 5'UTS to 5'UTS+ occur with probability $\rho$, transitions from TES to TTS- occur with probability $\zeta$, and all other allowed transitions have probability 1. The transition probabilities $\rho$ and $\zeta$ are estimated from the data, and are allowed to depend on the base sequence of the triplet; in addition, the probability $\rho$ also depends on the base sequence context around the triplet (*Kozak, 1987*). In this work, we assume that translation termination occurs at the first in-frame stop codon (*Equation 8*), i.e., we do not consider stop codon readthrough.

Conditional on the state assignments, we modeled 1) the total RPF abundance within a triplet, to account for the observation that translated base positions have a higher average RPF count compared to untranslated base positions, and 2) the distribution of RPF counts among the base positions in a triplet, to account for the periodicity in RPF counts within translated triplets. We explicitly accounted for differences in RPF abundance due to differences in transcript expression levels by using transcript-level RNA-seq data as a normalization factor. The short lengths of ribosome footprints mean that many base positions are unmappable; we treated triplets with unmappable positions by modifying the emission probabilities accordingly. Finally, we accounted for the additional variation in RPF counts across triplets assigned to the same state by modeling overdispersion in the triplet RPF abundance (see Materials and methods for details).

To quantify the accuracy of our model, we designed a simulation scheme to estimate what fraction of our inferred translated sequences are false discoveries. We first estimated the Type 1 error rate – i.e., the probability of inferring a translated region when no such region exists – using a set of simulated transcripts that had no signal of translation (null transcripts). The simulated transcripts were constructed by permuting the observed footprint counts in annotated coding transcripts. We then used this estimate to quantify the false discovery rate for each class of translated CDSs identified by riboHMM. Independently, using a simulated set of transcripts containing some signal of translation, we quantified the proportion of transcripts where our model incorrectly identified the precise translation initiation site conditional on having identified a translated sequence (see Materials and methods for details on the simulations).

## Results

### Application to human lymphoblastoid cell lines

We applied riboHMM to infer translated CDSs in human lymphoblastoid cell lines (LCLs) for which gene expression phenotypes were measured genome-wide: mRNA in 86 individuals, ribosome occupancy in 72 individuals and protein levels in 60 individuals (*Lappalainen et al., 2013*; *Battle et al., 2015*). We first assembled over 2.8 billion RNA sequencing reads into transcripts using StringTie (*Pertea et al., 2015*). This assembly gives us annotated transcripts that are expressed in LCLs, along with novel transcripts that do not overlap any GENCODE annotated gene. (We do not consider novel isoforms of annotated genes in our analyses.) Restricting to transcripts with at least five footprints mapped to each exon, we used riboHMM to identify high-confidence translated CDS. We learned the maximum likelihood estimates of the model parameters using the top five thousand highly expressed genes. The estimated parameters are robust to the choice of the learning set (*Figure 1—figure supplements 1* and *2*). Using these parameters, we then inferred the maximum *a posteriori* (MAP) frame and latent state sequence for each of the assembled transcripts. We retained transcripts whose MAP frame and state sequence corresponded to a pair of translation initiation and termination sites and had a joint posterior probability greater than 0.8. Using a set of simulated null transcripts, we estimated that this posterior cutoff corresponded to a Type 1 error rate of 4.5% per transcript. The MAP frame and state sequence directly give us the nucleotide sequence with the strongest signal of translation; we refer to these as main coding sequences or mCDS.

### Detection of novel CDSs in LCLs

Among 7801 GENCODE annotated coding genes for which we could infer a high posterior mCDS, we recovered the annotated reading frame for at least one transcript isoform in 7491 genes (96%); of these, we recovered the exact annotated CDS in 4500 genes. In the remaining 310 genes, among

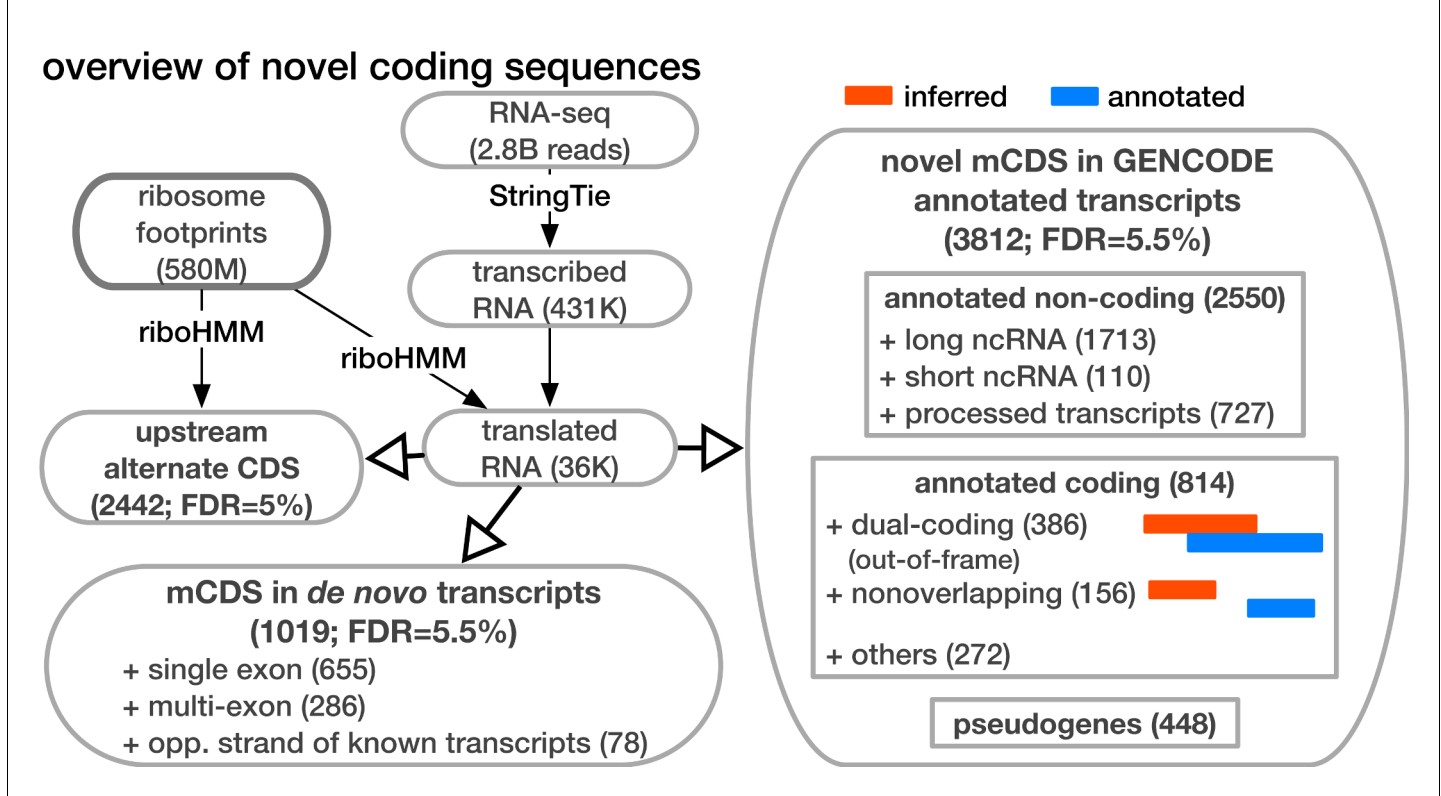

**Figure 2.** Overview of novel coding sequences. The analysis workflow indicates the size of the data (in read/footprint depth, or number of transcripts) at each step and the numbers and classes of transcript within which novel translated sequences were identified. Transcripts assembled by StringTie that do not overlap any annotated gene are called 'novel transcripts'. Long non-coding RNA includes lincRNAs, antisense transcripts and transcripts with retained introns, short non-coding RNA includes snRNA, snoRNA and miRNA, processed transcripts are transcripts without a long, canonical ORF, and pseudogenes include all subclasses of such genes annotated by GENCODE.

The following figure supplement is available for figure 2:

**Figure supplement 1.** Decision rules to identify novel mCDS.

all isoforms where we inferred an mCDS, the mCDS had a distinct reading frame from the annotated CDS (*Figure 1—figure supplement 3* details the rules that decide how our inference agrees with GENCODE). Of all GENCODE coding genes, we identified 814 GENCODE isoforms where our method identified an mCDS with a distinct reading frame from the annotated CDS. This set of 814 includes both isoforms within the 310 genes and additional isoforms within the 7491 genes (i.e., excluding the isoforms where the mCDS matched the frame of the annotated CDS).

We used simulations to estimate the accuracy of our inferences. For transcripts that do contain a translated sequence, we find that riboHMM inaccurately identifies an overlapping, translated sequence in a different frame at extremely low rates (Type 1 error rate = 0.31%). In contrast, riboHMM has a relatively high error rate for identifying the precise translation initiation site (false discovery proportion = 38%; see Materials and methods for details). Among transcripts where riboHMM correctly identified the reading frame, the concordance between the inferred and annotated translation initiation site does not correlate with the length of CDS (Mann-Whitney test; p-value = 0.12). Amongst these, when riboHMM did not identify the annotated initiation site, the inferred initiation site was equally likely to be upstream or downstream of the annotated initiation site (Mann-Whitney test; p-value = 0.41). Our analysis is robust to sequencing depth; *Figure 1—figure supplement 4* illustrates that nearly 60% of annotated coding sequences identified with the full data set (580 million footprints) could be accurately recovered even when the sequencing depth was reduced by almost two orders of magnitude.

Thus, in summary, it is likely that most of the 814 mCDS that were identified within GENCODE annotated protein-coding transcripts and have a distinct reading frame compared to GENCODE annotations are indeed novel alternate translated sequences. To ensure that an mCDS is truly novel, we verified that it does not overlap any known CDS annotated by GENCODE, UCSC (*Rosenbloom et al., 2015*), or CCDS (*Farrell et al., 2014*) in the same frame. (See *Figure 2* for the different classes of LCL transcripts that contain a novel mCDS; *Figure 2—figure supplement 1* illustrates the decision rules used to identify a novel mCDS). Among these 814 novel mCDS, 386 mCDS overlap an annotated CDS but have a different reading frame (labeled 'dual-coding') and 156 do not overlap the annotated CDS. An example of a novel dual-coding region – an mRNA sequence that codes for proteins in two different frames – inferred in the POLR2M gene is illustrated in *Figure 3A*. Using tandem mass-spectrometry data (*Battle et al., 2015*), we found four unique spectra matching peptides in the mCDS and no spectra matching peptides in the annotated CDS (protein level posterior error probability = $3 \times 10^{-35}$). However, our simulations suggest that most, or all, of the 39% of genes where riboHMM infers the annotated reading frame but disagrees with the annotated start site are false discoveries, and these are not considered further here.

In addition, we identified 2550 novel mCDS in annotated non-coding transcripts and 1019 mCDS within novel transcripts assembled *de novo* by StringTie (FDR = 5.6%). Using simulations, we estimated that given a transcript has no translated sequence; riboHMM inaccurately identifies a translated sequence at fairly low error rates (Type I error rate = 4.5%). Over 60% of the mCDS in novel transcripts were identified in single-exon transcripts transcribed from regions containing no annotated genes, while about 8% were identified in novel transcripts transcribed from the strand opposite to an annotated transcript. Finally, we inferred mCDS in 448 expressed pseudogenes, among 14,065 pseudogenes annotated in humans (*Pei et al., 2012*); nearly 90% of these mCDS were identified in processed pseudogenes. An mCDS in pseudogene GAPDHP72 is shown in *Figure 3—figure supplement 1*, comparing the ribosome abundance and peptide matches to the pseudogene mCDS with those of its parent gene GAPDH.

Unlike current CDS annotations, which almost exclusively start at the methionine codon AUG, these novel mCDS taken together have a substantially higher usage of non-canonical codons, particularly CUG (*Figure 3B*), consistent with recent observations in mouse embryonic stem cells (*Ingolia et al., 2011*) and human embryonic kidney cells (*Lee et al., 2012*). This is despite the fact that we inferred the initiation site by assuming shared properties between novel and annotated CDS. Although riboHMM has a high error rate when identifying translation initiation sites, our use of a hierarchical model for the initiation sites suggests that the errors in our inferred start codons are likely to be unbiased. These novel mCDS are also significantly shorter than annotated CDSs (median lengths 23 vs. 339 amino acids, Mann-Whitney test p-value < $2.2 \times 10^{-16}$; *Figure 3C*). The overall amino acid content within novel mCDS is comparable to that within annotated CDS, with a slight enrichment for arginine, alanine, cysteine, glycine, proline, and tryptophan residues (binomial test, p-value < $1.1 \times 10^{-16}$; *Figure 3—figure supplement 2*).

Below, using an alternative measure of ribosome occupancy, we first assess independent evidence for translation initiation at many of these novel mCDS. Then, we test whether these mCDS are functional both using human polymorphism data and using substitution patterns across vertebrates. Finally, we characterize those mCDS whose peptide products were identified in mass-spectrometry data.

## Translation at novel mCDS validated using harringtonine-treated ribosome footprints

We next sought to provide independent experimental validation for the novel mCDS. A direct approach to validate translation initiation sites is to assay ribosome occupancy in cells treated with harringtonine (*Ingolia et al., 2011*). Harringtonine interacts with and arrests the initiation complex while leaving the elongation complex to continue translating and run off the transcript. Harringtonine-treated ribosome footprint profiling data therefore show a specific enrichment pattern at the translation initiation site; this pattern has previously been used to identify translation initiation sites in mouse embryonic stem cells (*Ingolia et al., 2011*). We measured harringtonine-treated ribosome footprints in two LCLs and aggregated the counts of footprints across all novel mCDS. We observed an enrichment of footprints at the inferred initiation site of the novel mCDS (binomial test, p-value = $9.5 \times 10^{-79}$; *Figure 4*), similar to the enrichment of aggregate ribosome occupancy at the

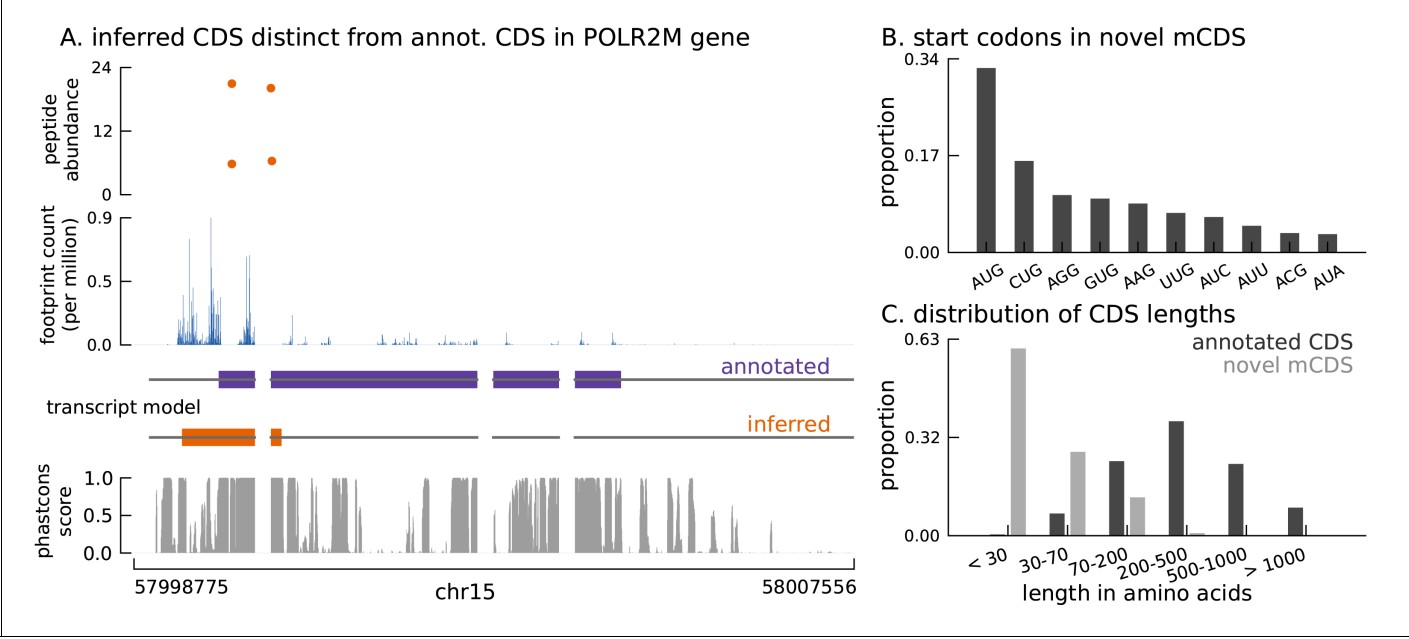

**Figure 3.** Thousands of novel translated sequences identified in annotated and novel transcript isoforms. (**A**) The inferred CDS for an isoform of the POLR2M gene overlaps its annotated CDS and is in a different frame. All four distinct peptides uniquely mapping to this gene match the inferred CDS (protein-level posterior error probability = $3 \times 10^{-35}$). (The introns and the last exon have been shortened for better visualization.) (**B**) Distribution of start codon usage across all novel mCDS. (**C**) Distribution of the lengths of the novel mCDS (gray) compared with the lengths of GENCODE annotated CDSs (black).

The following figure supplements are available for figure 3:

**Figure supplement 1.** Translated coding sequences identified in hundreds of pseudogenes.

**Figure supplement 2.** Comparing the amino acid content between annotated and novel CDS.

**Figure supplement 3.** Characteristics of peptides matched to novel CDS.

**Figure supplement 4.** Annotated genes with peptide hits tend to be longer, have higher expression and a distinct amino acid composition.

initiation sites of a matched number of mCDS that agreed exactly with the annotated CDS (see *Figure 4—figure supplement 1* for mCDS in pseudogenes). We observed a significant enrichment at both AUG (p-value = $5.2 \times 10^{-79}$) and non-AUG (p-value = $9.4 \times 10^{-25}$) initiation sites. The reduced enrichment for the novel mCDS compared to annotated CDSs is likely due to the lower levels of translation of these novel mCDS and the high error rate in identifying the precise base at which translation is initiated. Accounting for these limitations, our observation of enrichment suggests that ribosomes do initiate the translation of many of the novel mCDS identified by riboHMM.

## Selective constraint on coding function in novel mCDS

We next ascertained the functional importance of these novel mCDS based on the selective constraint imposed on random mutations that occur within them. A bi-allelic single nucleotide polymorphism (SNP) that falls within an mCDS can be inferred as synonymous or nonsynonymous depending on whether switching between the two alleles of the SNP changes the amino acid sequence of the mCDS. If the mCDS do not produce proteins that are functionally important, we expect the two classes of variants to have similar selection pressures on average, and thus to segregate at similar frequencies. Only if the novel mCDS produce functionally important peptides do we expect inferred nonsynonymous SNPs to segregate at lower frequencies than inferred synonymous SNPs.

Starting with biallelic SNPs identified using whole genome sequences of 2504 individuals (*Auton et al., 2015*), we examined the set of SNPs falling within all novel mCDS (13,907 variants

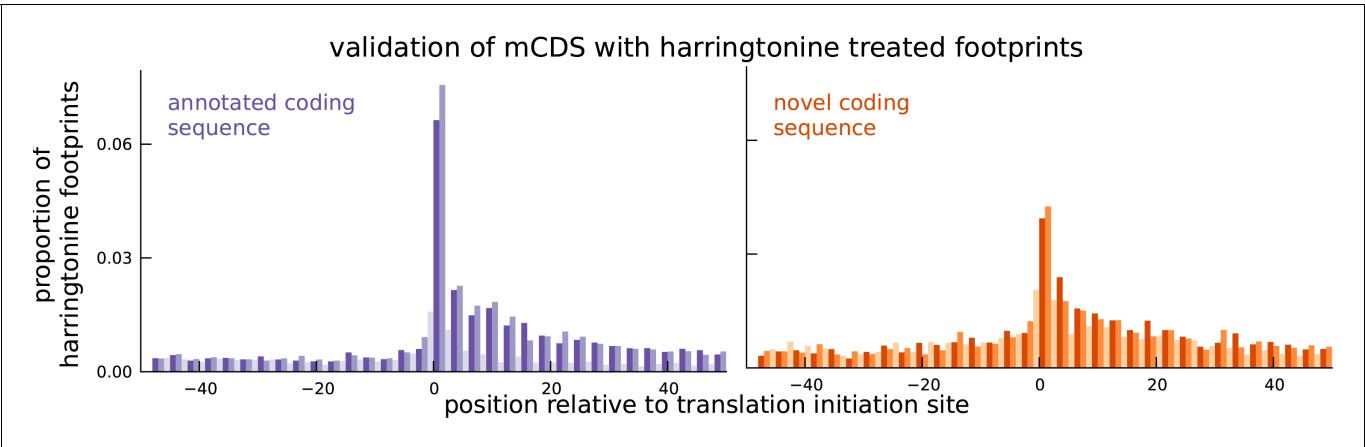

**Figure 4.** Validation of novel mCDS using harringtonine-treated ribosome profiling data. Harringtonine-treated ribosome footprints show enrichment at the inferred translation initiation sites, when aggregated across novel mCDS (orange), similar to the enrichment at the initiation sites of a matched number of mCDS that agreed exactly with the annotated CDS (purple), suggesting that ribosomes do initiate translation of the novel mCDS.

The following figure supplement is available for figure 4:

**Figure supplement 1.** Validation of translated sequences identified in pseudogenes.

within 3096 novel mCDS). We labeled each SNP as synonymous or nonsynonymous with respect to the inferred CDS and show the cumulative distribution of minor allele frequencies (MAF) of each SNP class (*Figure 5A*). We observed that nonsynonymous SNPs have an excess of rare variants compared with synonymous SNPs (Mann-Whitney test; p-value = $1.08 \times 10^{-4}$), implying a difference in the intensity of purifying selection (*Nielsen, 2005*). This observed excess suggests that the novel mCDS are under significant constraint, consistent with functional peptides, albeit weaker than at annotated CDS. The mCDS identified within pseudogenes alone also showed a similar excess of rare variants among nonsynonymous SNPs (Mann-Whitney test; p-value = $5.6 \times 10^{-3}$). Such an excess was not observed for pseudogenes that had detectable ribosome occupancy but lacked a high-confidence inferred coding sequence (*Figure 5—figure supplement 1*); for these pseudogenes, the SNPs were labeled based on the reading frame of the parent gene. This highlights that ribosome occupancy alone is insufficient to identify translated sequences, and our method is able to leverage finer scale structure in ribosome footprint data to detect functional coding sequences.

While the allele frequency spectra provide evidence that some of the novel mCDS are functional in present-day human populations, they are less informative about the long-term selective constraint on these sequences. To identify whether the novel mCDS have been under long-term functional constraint, we compared the substitution rates at synonymous and nonsynonymous sites within the novel mCDS using whole-genome multiple sequence alignments across 100 vertebrates. (We excluded mCDS identified in pseudogenes from this analysis due to difficulties in assigning orthology.) In *Figure 5B*, 232 novel mCDS have a significantly lower nonsynonymous substitution rate ($dN$) compared to their synonymous substitution rate ($dS$) after Bonferroni correction (p-value $< 2.91 \times 10^{-5}$), suggesting that these mCDS have been under long-term purifying selection. Since the power to detect significantly low values of dN/dS depends on the length of the CDS and the qualities of the genome assemblies and the multiple sequence alignments across distant species at these sequences, the number of functional novel CDSs identified is a conservative lower bound.

## Detection of novel proteins by mass spectrometry

We next tested whether we could detect the novel mCDS predictions using mass spectrometry data. We used a large data set of SILAC-labeled tandem mass-spectra generated by trypsin-cleavage of large, stable proteins in many of the same LCLs (*Battle et al., 2015*). Running MaxQuant (*Cox and Mann, 2008*) against the sequence database of 4831 novel mCDS, at 10% FDR, we identified 161 novel mCDS sequences that have at least one unique peptide hit – a tryptic peptide that matches a

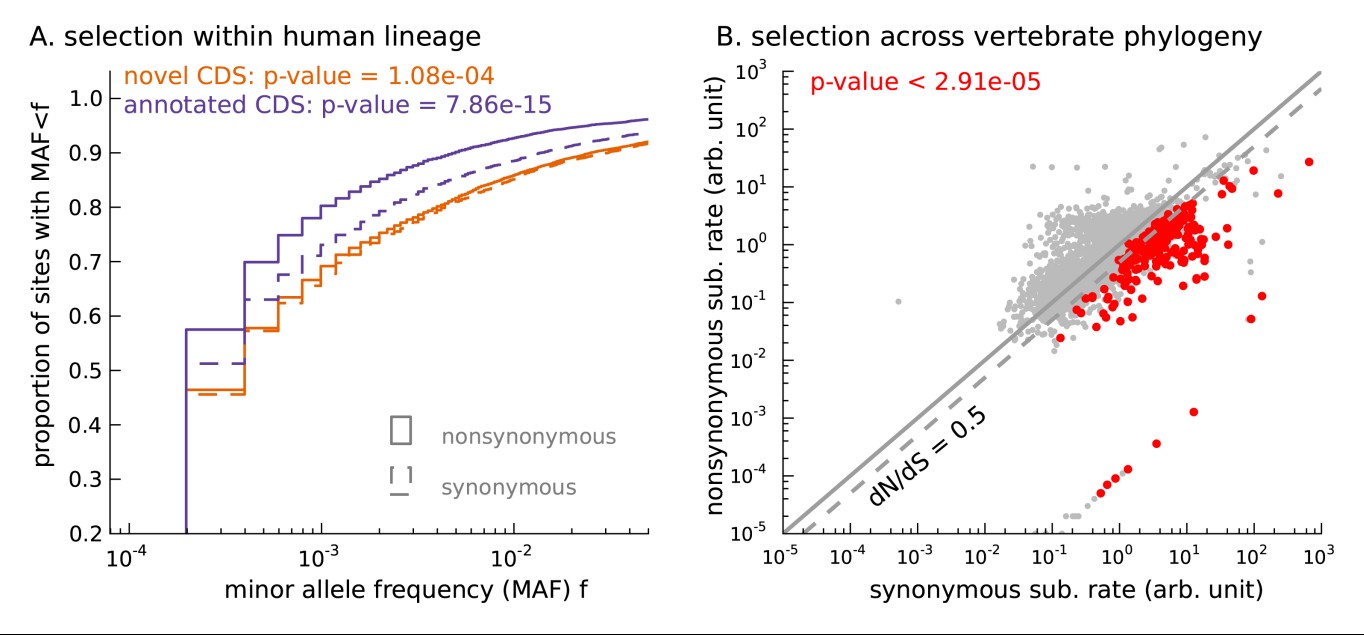

**Figure 5.** Novel translated sequences show significant signatures of coding function. (**A**) Genetic variants that are nonsynonymous with respect to the inferred mCDS segregate at significantly lower frequencies in human populations than synonymous variants. The novel regions are under weaker selective constraint compared to known CDS. (The numbers of variants in each class are matched between novel and annotated CDS.) (**B**) Scatter plot comparing the substitution rate at inferred synonymous variants versus inferred nonsynonymous variants for each novel mCDS, computed using multiple sequence alignments across 100 vertebrate species. Highlighted in red are 232 novel mCDS identified to be under significant long-term purifying selection after Bonferroni correction (testing for dN/dS < 1;), indicating conserved coding function for these sequences.

The following figure supplement is available for figure 5:

**Figure supplement 1.** Signature of coding function in translated sequences identified in pseudogenes.

mass-spectrum (*Supplementary file 1*). More than 70% of novel mCDS with a peptide hit have at least 2 distinct peptides matched to it and, in almost all cases, the unique peptides were independently identified in two or more LCLs (*Figure 3—figure supplement 3*).

To assess how many hits we would expect to the novel mCDS if their properties were like those of annotated CDSs, we developed a model that predicts whether an annotated protein has at least one mass-spectrum match, using features based on expression and sequence composition of the protein (see Materials and methods for more details). The mass-spectrometry data are highly biased towards detection of larger and more highly expressed proteins. Furthermore, the trypsin cleavage step of the experimental protocol imposes strong constraints on the set of unique peptide sequences that can be observed in an experiment. Assuming that the distributions of these predictive features estimated from annotated CDSs can be applied to the novel mCDS, we computed the expected number of novel mCDS with a peptide hit to be 603.

We thus find many fewer mass spectrometry hits to the novel mCDS than expected from a model calibrated on previously annotated mCDS (161 vs. 603). Since our model accounts for translation levels of the mCDS, the low validation rate is unlikely to be due to low rates of protein production. One possible explanation for the low validation rate may be that a large number of the inferred novel mCDS are false discoveries. However, our simulations highlight that our method has a low false positive rate and the Harringtonine data argue that many of the novel mCDS are correct predictions, thus we suggest that some other property of the mCDS may explain their low detection rate. In particular, it is possible that the novel proteins may have higher turnover rates than annotated proteins. For example it is possible that the proteins translated from novel mCDS may have substantially lower half-life than annotated proteins, or may be secreted, and thus have too low concentrations within the cell to be detectable by mass spectrometry assays.

## Translation of short alternate coding sequences in addition to the mCDS

Protein-coding transcripts in eukaryotes are typically annotated to have only one CDS (i.e., they are monocistronic). However, a number of studies have demonstrated that ribosomes can initiate translation at alternative start codons (*Xu et al., 2010*; *Ingolia et al., 2011*; *Lee et al., 2012*) and many others have identified transcripts with alternative CDSs encoding functional peptides (*Vanderperre et al., 2013*; *Kochetov, 2008*; *Barbosa and Romão, 2013*). Furthermore, anecdotal evidence has suggested that translation of the alternate CDS serves as a mechanism to suppress translation of the main CDS (*Lee et al., 2002*; *Hernández-Sánchez et al., 2003*; *Lammich et al., 2004*). However, assessing such a mechanism genome-wide has been challenging, mainly due to a lack of appropriate annotations (*Calvo et al., 2009*).

To this end, we adapted our approach to identify additional coding sequences within transcripts that are translated in LCLs. Assuming that the sub-codon structure of footprint abundance is similar between the main and alternate CDS, we identified 2442 novel CDSs upstream of the mCDS inferred by our method (FDR = 5%); we call them upstream alternate coding sequences or uaCDS (see Materials and methods for details; see also *Figure 6—figure supplement 1*). *Figure 6A* illustrates the ribosome footprint density within the uaCDS of the transmembrane gene TM7SF2, and its

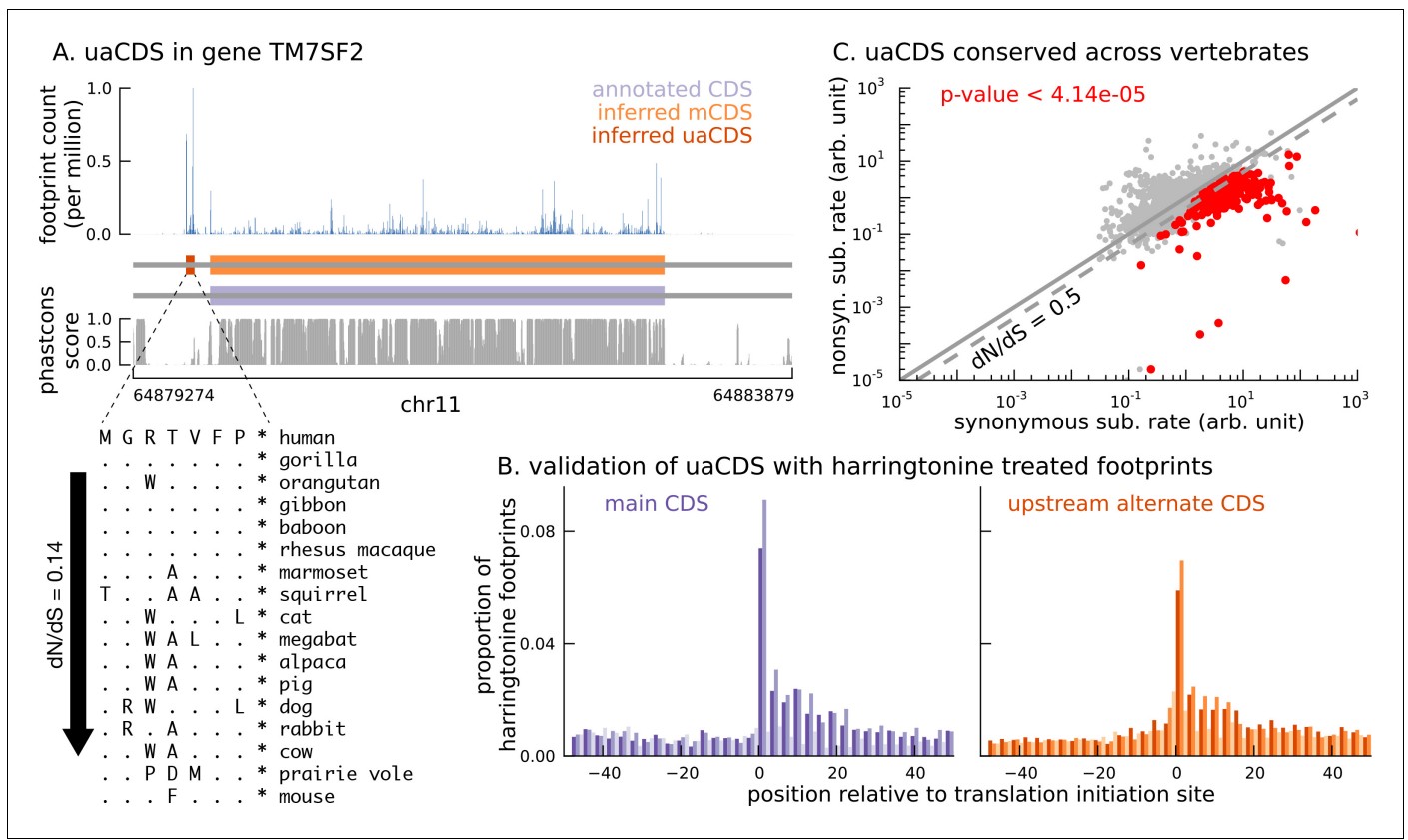

**Figure 6.** Short translated sequences identified upstream of thousands of translated main coding sequences. (**A**) An alternate, novel CDS was identified upstream of the inferred main CDS in gene TM7SF2. As shown in its protein sequence alignment across mammals, the uaCDS (in particular, the start and stop codons) is highly conserved with dN/dS = 0.14. (**B**) Harringtonine-treated ribosome footprints show strong enrichment at the inferred initiation sites of uaCDS, comparable to the enrichment at the initiation sites of the corresponding mCDS, suggesting that ribosomes do initiate translation of these uaCDS. (**C**) Using multiple sequence alignment across 100 vertebrate species, 317 uaCDS were identified to have strong, significant long-term conservation.

The following figure supplement is available for figure 6:

**Figure supplement 1.** Characteristics of novel uaCDS.

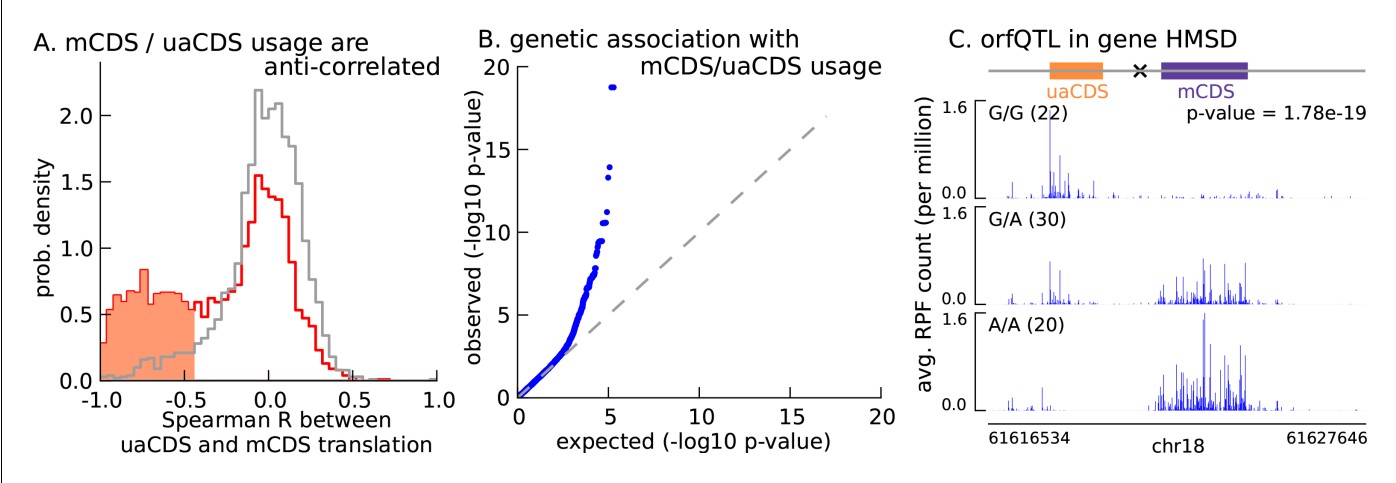

**Figure 7.** Translation of uaCDS regulates translation of mCDS. (**A**) Spearman correlation, across LCLs, between mCDS translation and uaCDS translation (red histogram). Using random (mCDS, uaCDS) pairs, matched for length and pairwise distance, we computed an empirical null distribution of Spearman correlations (gray histogram). At 10% FDR, 917 inferred (uaCDS, mCDS) pairs have significant negative correlation (shaded red region). (**B**) 365 orfQTLs (genetic variants associated with ORF usage; i.e., whether the mCDS or uaCDS of a transcript is translated) were identified at 10% FDR (41 pairs of mCDS/uaCDS). (**C**) Illustrating an example of an orfQTL in the histocompatibility minor serpin domain-containing (HMSD) gene (introns removed for better visualization). The most significant variant (marked x) lies within an intron between the mCDS and uaCDS of the transcript.

conservation across mammals. We find strong enrichment of harringtonine-treated ribosome footprints at the initiation sites of uaCDS similar to the initiation sites of mCDS in the same transcripts (*Figure 6B*). Using mass-spectrometry data, we identified 46 uaCDS that have at least one peptide hit, substantially lower than the expectation of 891 hits predicted by our model. Finally, comparing the substitution rates at inferred synonymous and nonsynonymous sites, we identified 317 uaCDS with highly constrained coding function (*Figure 6C*). Those uaCDS with a peptide match and those having evidence of constrained coding function are not concordant (Fisher's test; p-value = 0.56), consistent with the low sensitivity of standard mass-spectrometry protocols to identify very short proteins.

## Translation of uaCDS negatively correlates with translation of mCDS

With 2442 uaCDS identified as translated in LCLs, we next tested the hypothesis that uaCDS expression negatively correlates with mCDS for each pair. We observed that, at 10% FDR, 917 pairs of uaCDS and mCDS had significant negative correlations across individuals between the proportion of footprints assigned to them (*Figure 7A*). Our observation that nearly 40% of pairs of uaCDS and mCDS are significantly anti-correlated, despite incomplete power due to low sample size, suggests that a potential role of alternate CDSs in a transcript is to regulate the translation of the main CDS.

Variation in ORF usage can be driven by a number of factors including *cis* genetic effects and *trans* effects like variation in expression of RNA binding proteins. To identify *cis* variants that affect ORF usage in a bicistronic transcript, we tested for association of the proportion of RPFs assigned to the mCDS (or uaCDS) with variants in a 10-kilobase window around the transcript; this phenotype effectively controls for variation in gene expression across the LCLs. We identified 365 *cis* orfQTLs (genetic variants associated with ORF usage) across 41 pairs of mCDS and uaCDS at 10% FDR (*Figure 7B*). In *Figure 7C*, we illustrate an example of an orfQTL in a bicistronic transcript of the HMSD gene (histocompatibility minor serpin domain-containing); this gene is also known to have a distinct genetic variant associated with alternative usage of two coding isoforms (*Kawase et al., 2007*). Our observation of orfQTLs in a number of genes distinguishes ORF usage as an additional layer of post-transcriptional regulation of protein expression.

## Discussion

We developed riboHMM, a mixture of hidden Markov models to accurately resolve the precise set of mRNA sequences that are being translated in a given cell type, using sequenced RPFs from a ribosome profiling assay, sequenced reads from an RNA-seq assay and the RNA sequence. When applied to human LCLs, this method was able to accurately identify the translated frame in 96% of annotated coding genes that had a high posterior mCDS. In addition, a key advantage of our framework is the ability to infer novel translated sequences that may be missed by annotation pipelines that focus on long CDSs (>100 amino acids), conservation based approaches that require long-term purifying selection, or direct proteomics measurements that are biased toward highly expressed, stable proteins. We used riboHMM to identify 7273 novel CDSs, including 448 of novel translated sequences in pseudogenes and 2442 bicistronic transcripts that contain an upstream CDS in addition to a main CDS. We observed enrichment in harringtonine-arrested ribosome occupancy at the inferred translation initiation sites, suggesting that many of the novel mCDS are real. These novel sequences showed significant differences in the amount of purifying selection acting on inferred non-synonymous versus synonymous sites, suggesting that many of these sequences are conserved as functional peptides, including those mCDS identified in lncRNAs, pseudogenes and novel transcripts.

One caveat of our model is its restriction on one CDS per transcript. In this study, we worked around this limitation using a greedy approach and identified thousands of transcripts with multiple CDSs (either two non-overlapping inferred CDSs or an inferred mCDS distinct from the annotated CDS). Indeed, in some instances where the frame of the mCDS and annotated CDS of a transcript disagreed, we found strong support from mass-spec data for the inferred mCDS frame (*Figure 3A*). These observations highlight the existence of a large number of transcripts in humans that have multiple CDSs and the variation in alternative usage of CDSs across tissues, an area that has largely been overlooked. Additionally, riboHMM does not effectively distinguish footprints arising from different isoforms and, thus, cannot resolve overlapping translated sequences from multiple coding isoforms of a gene. Extending riboHMM to model multiple, possibly overlapping CDSs jointly across multiple isoforms could help uncover this additional layer of complexity in the human genome.

In addition to identifying individual novel coding sequences, our method enables us to observe general properties shared across these coding regions. Interestingly, we found novel coding sequences to have a higher usage of non-AUG start codons than would be expected by considering current translation initiation site annotation (*Figure 3B*). We emphasize that although our model assumes shared properties between novel CDS and annotated CDS, we did not use any information about annotated translation initiation and termination sites when learning the model parameters. We used well-expressed genes as our learning set to ensure that when the footprint data do not provide very strong evidence regarding the initiation site, novel coding sequences identified by our method are as similar as possible to annotated coding sequences in the sequence composition of their initiation sites. While this allows us to be conservative and identify novel CDS that are similar to annotated CDS in their ribosome footprint patterns, our approach will not be able to identify translation events that differ in their footprint patterns from the majority of translation events. In other words, our choice of learning set could potentially bias the inference. Nevertheless, similar start codon usage frequencies were observed when random sets of 5000 genes were used as learning set (*Figure 1— figure supplement 7*) further confirming the robustness of our method.

To improve our ability to identify the translation initiation site, we attempted to incorporate harringtonine treated data in the model by introducing an additional covariate in the transition probabilities, providing independent information on the positions of translation initiation sites. However, the codon usage at the inferred initiation sites showed no significant change (K-S test; p-value = 0.88) and the set of inferred coding sequences showed very little difference when harringtonine data were incorporated into the model. Since the footprint data without treatment show clear enrichment at initiation sites, it is likely that harringtonine treated data do not provide much additional information. Thus, while the harringtonine treated data were useful as independent validation for our inferred initiation sites, the data did not have sufficient additional information to calibrate the confidence in our predicted initiation sites for each transcript.

While the precise function of these novel CDSs remains unclear, we found evidence supporting a regulatory role for novel alternate CDSs identified upstream of the mCDS (uaCDS). Although it is

unclear whether the down regulation of mCDS by uaCDS is dependent on the peptide sequences of uaCDS, our finding is consistent with previous assertions under which translation of upstream ORFs regulates translation of the main CDS in cap-dependent translation initiation (*Morris and Geballe, 2000*).

Our method provides an alternative framework for annotating the coding elements of the genome. Compared to current methods that use sequence information in cDNA and protein data-bases and those that rely on high-quality genome annotations in closely related species, riboHMM provides a relatively unbiased CDS annotation and opportunities for finding entirely novel CDSs. In particular, one could use riboHMM to identify the set of CDS for a species within a poorly annotated evolutionary clade, using ribosome profiling and RNA seq data immediately after its genome is sequenced and assembled. In addition, given ribosome footprint profiling data from multiple cell types, riboHMM can be used to investigate cell-type-specific translation of coding elements beyond cell-type-specific gene or isoform expression. These features render this tool particularly useful in studying molecular evolution of newly arisen coding genes and linking tissue-specificity of CDS usage to disease.

## Materials and methods

### Assembling expressed transcripts in LCLs

We mapped paired-end 75 bp RNA-seq reads pooled across 85 Yoruba lymphoblastoid cell lines (*Lappalainen et al., 2013*) to the Genome Reference Consortium Human Reference 37 (GRCh37) assembly using STAR (*Dobin et al., 2013*), with the additional flag –outSAMstrandField intronMotif to aid transcript assembly downstream, resulting in 2.8 billion uniquely mapped fragments. Using the mapped reads, we assembled models of transcripts expressed in LCLs using StringTie v1.0.4 (*Pertea et al., 2015*), and used GENCODE v.19 transcript models to guide the assembly. In addition, we required that the lowest expressed isoform of a gene have no less than 1% the expression of the highest expressed isoform (-f 0.01), and that each exon-exon junction be supported by at least 2 spliced reads (-j 2). Since the RNA-seq protocol did not produce strand-specific reads, we treated the forward strand and reverse strand of a transcript model assembled by StringTie as distinct transcripts. Our final set of 430,754 expressed transcripts included 122,168 GENCODE annotated transcript isoforms and 308,586 novel isoforms. (We did not consider novel isoforms of annotated genes identified by StringTie.)

### Ribosome footprint profiling

Cell lines used in this study were ordered from Coriell Institute for Medical Research (https://www.coriell.org). To verify the identity of each cell line, we used genotype information derived from the sequencing data. To inspect potential contamination by mycoplasma, we used Universal Myco-plasma Detection Kit from ATCC (ATCC 30-1012K). Ribosome footprint profiling experiments and sequencing data processing were performed as previously described (*Battle et al., 2015*), with the exception of a harringtonine treatment step to arrest ribosomes at the sites of translation initiation. Briefly, lymphoblastoid cell lines, GM19204 and GM19238, were cultured at 37°C with 5% $CO_2$ in RPMI media with 15% FBS. The media were further supplemented with 2 mM L-glutamate, 100 IU/ml penicillin, and 100 μg/ml streptomycin. Right before cell lysate preparation, each culture was treated with 2 μg/ml harringtonine (final concentration in media) for 2 min followed by 100 μg/ml cycloheximide (final concentration in media). For ribosome profiling experiments, ARTseq Ribosome Profiling kit for mammalian cells (RPHMR12126) was used following vendor's instructions. Sephacryl S400 spin columns (GE; 27-5140-01) were used for monosome isolation. Libraries were sequenced on an Illumina HiSeq 2500. For sequencing data processing and mapping, adaptor sequences were removed from the 3' end of each read using the Clipper tool from the FASTX-Toolkit. In addition, the 5' most nucleotide (commonly resulted from non-templated additions) was removed using the Trimmer tool from the FASTX-Toolkit. To increase mapping efficiency, we filtered out sequence reads that mapped to rRNA, tRNA or snoRNA (FASTA files downloaded from Ensembl on 05/02/13) using Bowtie 2, version 2.0.2 (*Langmead and Salzberg, 2012*). Processed reads were aligned to genome build hg19 (human) using TopHat v2.0.6 (*Trapnell et al., 2009*). The mapping step was guided by transcriptome annotations (downloaded from Ensembl on 01/31/13).

## Mixture of HMMs to model translated coding sequences

Consider $N$ transcripts where the $n^{th}$ transcript has length of $L_n$ assumed to be a multiple of three ($L_n = 3M_n$; see *Transcripts with length not a multiple of three* for details on how our model handles the remaining one or two base positions when $L_n$ is not a multiple of three). Our data consist of RPF counts $T = (T^n)_{n=1}^N$, RNA sequence $S = (S^n)_{n=1}^N$, and transcript expression $E = (E^n)_{n=1}^N$ (in units of RNA-seq reads per base position per million sequenced reads) on $N$ transcripts, where $T^n$ and $S^n$ are vector quantities and $E^n$ is a scalar aggregated over the entire length of the transcript. Let $T^n = \left(T_1^n, \cdots, T_{L_n}^n\right)$ and $S^n = \left(S_1^n, \cdots, S_{L_n}^n\right)$, where $T_b^n$ and $S_b^n$ denote the RPF counts and the base at the $b^{th}$ position in the $n^{th}$ transcript, respectively. We model the footprint data $T$ using a mixture of HMMs that incorporates $S$ and $E$. Assuming independence across transcripts, the probability of $T$ given $S$ and $E$ is written as $P(T|\Theta, S, E) = \sum_{n=1}^N P(T^n|\Theta, S^n, E^n)$ where $\Theta$ denotes the set of model parameters.

### Mixture of three reading frames for a transcript

To capture the three-base structure in RPF data within the CDS, we represent each transcript as a sequence of non-overlapping base triplets, some of which potentially represent codons. Since the CDS of the transcript could belong to one of three reading frames (as illustrated in *Figure 1B*), we introduced a latent frame variable, $F^n \in \{1, 2, 3\}$, that specifies the reading frame for the $n^{th}$ transcript. Then, given $F^n = f$, $T^n$ can be represented as a sequence of $M_n - 1$ triplets and three remaining base positions (see *Figure 1B*). Specifically, $T^n|F^n = f := \left(X_{f,1}^n, \cdots, X_{f,(M_n-1)}^n, R_f^n\right)$, where $X_{f,m}^n = \left(T_{3m-3+f}^n, T_{3m-2+f}^n, T_{3m-1+f}^n\right)$ and

$$
R_f^n = \begin{cases} \left(T_{L_n-2}^n, T_{L_n-1}^n, T_{L_n}^n\right) & \text{if } f = 1 \\ \left(T_1^n, T_{L_n-1}^n, T_{L_n}^n\right) & \text{if } f = 2 \\ \left(T_1^n, T_2^n, T_{L_n}^n\right) & \text{if } f = 3 \end{cases} \tag{1}
$$

The probability of $T^n$ is then given by

$$
\begin{aligned}
P(T^n|\Theta, S^n, E^n) &= \sum_{f=1}^3 P(T^n|F^n = f, \Theta, S^n, E^n) P(F^n = f|\Theta, S^n, E^n) \\
&= \sum_{f=1}^3 P\left(X_{f,1}^n, \cdots, X_{f,(M_n-1)}^n, R_f^n|F^n = f, \Theta, S^n, E^n\right) P(F^n = f|\Theta, S^n, E^n)
\end{aligned} \tag{2}
$$

We assumed that the probability over $F^n$ is independent of $S^n$ and $E^n$, and is uniform over all three frames, $P(F^n = f|\Theta, S^n, E^n) = \frac{1}{3}$. In addition, we assumed that the RPF data from the sequence of triplets and the RPF data from the three remaining base positions are independent, leading to

$$
P\left(X_{f,1}^n, \cdots, X_{f,(M_n-1)}^n, R_f^n|F^n = f\right) = P\left(X_{f,1}^n, \cdots, X_{f,(M_n-1)}^n|F^n = f\right) P\left(R_f^n|F^n = f\right). \tag{3}
$$

(For notation convenience, we have dropped highlighting the dependence of $X^n$ and $R^n$ on $\Theta$, $S_n$, and $E_n$.) We modeled the probability of the data from the sequence of triplets, $P\left(X_{f,1}^n, \cdots, X_{f,(M_n-1)}^n|F^n = f\right)$, using an HMM, and the probability of the data from the remaining positions, $P\left(R_f^n|F^n = f\right)$, using a Poisson-gamma model as described below.

### HMM for each frame of a transcript

The pattern of RPF count data in triplets depends on whether the triplet is being translated or not. To model these patterns, we assumed that each triplet belongs to one of nine states (see *Figure 1B*): 5' Untranslated State (5'UTS), last untranslated triplet 5' to the CDS (5'UTS+), Translation Initiation State (TIS), state after TIS (TIS+), Translation Elongation State (TES), state before TTS (TTS-), Translation Termination State (TTS), first untranslated triplet 3' to the CDS (3'UTS-), and 3' Untranslated State (3'UTS). The five states (TIS+, TIS, TES, TTS-, TTS) denote translated triplets and

the remaining four states (5'UTS, 5'UTS+, 3'UTS-, 3'UTS) denote untranslated triplets. In particular, the start codon corresponds to the base triplet assigned to the TIS state and the stop codon corresponds to the base triplet assigned to the 3'UTS- state. The groups of states (5'UTS+, TIS, TIS+) and (TTS-, TTS, 3'UTS-) help model the distinct features of the footprint data around the translation initiation and termination sites, respectively. We introduced a sequence of $M_n - 1$ hidden variables $Z_f^n = \left( Z_{f,\,1}^n, \cdots, Z_{f,\,(M_n-1)}^n \right)$ for each frame of the $n^{th}$ transcript, where $Z_{f,\,m}^n$ denotes the state for the $m^{th}$ triplet in the $f^{th}$ frame.

For each state, an emission probability for $X_{f,\,m}^n$ can be modeled as follows. Let $Y_{f,\,m}^n$ denote the sum of three elements in $X_{f,\,m}^n$ (i.e., the total RPF count for the $m^{th}$ triplet). Then, $\mathrm{P}\left( X_{f,\,m}^n | Z_{f,\,m}^n = z \right) = \mathrm{P}\left( X_{f,\,m}^n | Y_{f,\,m}^n, Z_{f,\,m}^n = z \right) \mathrm{P}\left( Y_{f,\,m}^n | Z_{f,\,m}^n = z \right)$ and

$$X_{f,m}^n | Y_{f,m}^n, Z_{f,m}^n = z \sim \mathrm{multinomial}\left( Y_{f,m}^n, \pi_z \right), \tag{4}$$

$$Y_{f,m}^n | Z_{f,m}^n = z \sim \mathrm{Poisson}\left( \mu_{zfm}^n E^n \right), \tag{5}$$

$$\mu_{zfm}^n \sim \mathrm{gamma}\left( \alpha_z, \beta_z \right), \tag{6}$$

where the density of the gamma distribution is $\mathrm{P}(\mu) = \frac{\beta^{\alpha\beta}}{\Gamma(\alpha\beta)} \mu^{\alpha\beta-1} \exp^{-\beta\mu}$ with the mean and variance equal to $\alpha$ and $\frac{\alpha}{\beta}$, respectively.

The periodicity of RPF counts within the CDS is captured by the multinomial distribution with parameters $\pi_z = \left( \pi_{z,\,1}, \pi_{z,\,2}, \pi_{z,\,3} \right)$, where we assume $\pi_z = \left( \frac{1}{3}, \frac{1}{3}, \frac{1}{3} \right)$ for $z \in \{ 5'\mathrm{UTS},\ 5'\mathrm{UTS}+,\ 3'\mathrm{UTS}-,\ 3'\mathrm{UTS} \}$ to capture the lack of periodicity in the RPF data in untranslated regions. Furthermore, we allow the pattern of periodicity to differ across five states (TIS, TIS+, TTS, TTS-, TES).

The Poisson distribution for $Y_{f,\,m}^n$ captures the difference in RPF abundance between translated and untranslated regions (precisely, difference in abundance between triplets in different states). We corrected for differences in RPF abundance across transcripts due to differences in transcript expression levels by using $E^n$ as a transcript-specific normalization factor (see *Figure 1—figure supplement 5*). To account for additional variation in the RPF counts across triplets in the same state (e.g., due to varying translation rates across transcripts, and translational pausing), we allowed for triplet-specific parameters $\mu_{zfm}^n$ in the Poisson intensity and assumed that those parameters follow a gamma distribution. Under this model, $\mathrm{E}\left[ Y_{f,\,m}^n | Z_{f,\,m}^n = z \right] = \alpha_z E^n$ and $Var\left[ Y_{f,\,m}^n | Z_{f,\,m}^n = z \right] = \frac{\alpha_z}{\beta_z} E^{n^2} + \alpha_z E^n$.

We assumed that the sequence of hidden variables $Z_f^n$ follow a Markov chain. The assumption of up to one CDS in each transcript leads to a transition probability shown in *Figure 1B*, where $\rho_{f,\,m}^n = \mathrm{P}\left( Z_{f,\,m+1}^n = 5'\mathrm{UTS}+\ |\ Z_{f,\,m}^n = 5'\mathrm{UTS} \right)$ and $\zeta_{f,\,m}^n = \mathrm{P}\left( Z_{f,\,m+1}^n = \mathrm{TTS}-\ |\ Z_{f,\,m}^n = \mathrm{TES} \right)$ depend on the underlying RNA sequence and are given by

$$\rho_{f,\,m}^n = \begin{cases} \mathrm{logistic}\left( \psi_\kappa K_{f,\,m+2}^n + \sum_{c\in\Omega_{\mathrm{start}}} \psi_c \mathbb{I}\left[ M_{f,\,m+2}^n = c \right] \right), & \mathit{if}\ M_{f,\,m+2}^n \in \Omega_{start} \\ 0, & \mathrm{otherwise} \end{cases} \tag{7}$$

$$\zeta_{f,m}^n = \begin{cases} 1, & \mathit{if}\ M_{f,\,m+3}^n \in \Omega_{stop} \\ 0, & \mathrm{otherwise} \end{cases} \tag{8}$$

where $\mathbb{I}[\cdot]$ is the indicator function, $M_{f,\,m}^n = \left( S_{3m-3+f}^n, S_{3m-2+f}^n, S_{3m-1+f}^n \right)$ denotes the base sequence of the $m^{th}$ triplet, and $K_{f,\,m}^n$ denotes the log of ratio of likelihood under the Kozak model to likelihood under a background model of the base sequence flanking the $m^{th}$ triplet (see *Kozak model* for details). In our analysis, $\Omega_{start}$ contained the canonical start codon and all near-cognates, $\Omega_{\mathrm{start}} = \{\mathrm{AUG, CUG, GUG, UUG, AAG, ACG}\}$ and $\Omega_{\mathrm{stop}}$ contained the canonical stop codons, $\Omega_{\mathrm{stop}} =$

{UAA, UAG, UGA}. The parameters, $\psi_c$ and $\psi_\kappa$, indicate the importance of the triplet base sequence and the flanking base sequence in determining transition from untranslated triplets to translated triplets. The current specification of $\zeta_{f,m}^n$ and $\Omega_{\text{stop}}$ forces the coding sequence to terminate at the first in-frame occurrence of a stop codon. This model can be extended to account for stop codon read-through by using a logistic function for $\zeta_{f,m}^n$ for the same set $\Omega_{\text{stop}}$.

## Model for $R_f^n$

We model $R_f^n$, the RPF counts at bases before or after the sequence of triplets (see *Equation 1*), using the emission probabilities of the $5'\text{UTS}$ or $3'\text{UTS}$ states. Assuming that the three elements of $R_f^n$ are independent, we have $\text{P}\left(R_f^n|F^n=f\right) = \prod_{i=1}^3 R_{f,i}^n|F^n=f$. Each element can be modeled as

$$R_{f,i}^n \sim \text{Poisson}\left(\frac{1}{3}\lambda_{fi}^n E^n\right), \tag{9}$$

$$\lambda_{fi}^n \sim \text{gamma}\left(\alpha_z,\beta_z\right), \tag{10}$$

where $z = 5'\text{UTS}$ if $R_{f,i}^n \in \left\{T_1^n, T_2^n\right\}$, and $z = 3'\text{UTS}$ if $R_{f,i}^n \in \left\{T_{L_n-2}^n, T_{L_n-1}^n, T_{L_n}^n\right\}$.

## Parameter estimation and inference

We used an EM algorithm to compute the maximum likelihood estimate for the model parameters $\Theta = \left\{\pi_z,\alpha_z,\beta_z,\psi_\kappa,\psi_c\right\}$, that is, $\hat{\Theta} := \text{argmax}_\Theta \text{P}(T|\Theta, S, E)$.

To infer the translated CDS for the $n^{th}$ transcript, we identified the frame and state sequence that maximizes the joint posterior probability

$$\left(z^{n^*},f^{n^*}\right) := \text{argmax}_{zf}\text{P}\left(Z_f^n=z,F^n=f|T^n,S^n,E^n,\hat{\Theta}\right). \tag{11}$$

We first computed the maximum *a posteriori* (MAP) state sequence for each reading frame using the Viterbi algorithm, $z_f^{n^*} := \text{argmax}_z\text{P}\left(Z_f^n=z|F^n=f,T^n,S^n,E^n,\hat{\Theta}\right)$ for $f = 1, 2, 3$. Then, the MAP state sequence and frame is given as

$$\left(z^{n^*},f^{n^*}\right) := \text{argmax}_f\text{P}\left(Z_f^n=z_f^{n^*}|F^n=f,T^n,S^n,E^n,\hat{\Theta}\right)\text{P}\left(F^n=f|T^n,S^n,E^n,\hat{\Theta}\right), \tag{12}$$

where $z_f^{n^*}$ is a function of $f$, $\text{P}\left(F^n=f|T^n,S^n,E^n,\hat{\Theta}\right) \propto \text{P}\left(T^n|F^n=f,S^n,E^n,\hat{\Theta}\right)\text{P}\left(F^n=f\right)$ and $\text{P}\left(T^n|F^n=f,S^n,E^n,\hat{\Theta}\right)$ is the probability of the data marginalized over the latent states.

In our analyses, we estimated the model parameters using the top five thousand highly expressed genes. Then, we inferred the translated CDS for those transcripts in which each exon has at least five distinct ribosome footprints mapping to it. We restricted our further analyses to transcripts where (1) $\text{P}\left(Z_f^n=z^{n^*},F^n=f^{n^*}|T^n,S^n,E^n,\hat{\Theta}\right)>0.8$, (2) the MAP state sequence $z^{n^*}$ contains a TIS state and a TTS state (i.e., a pair of initiation and termination sites), (3) more than 50% of base positions within the inferred CDS are mappable, and (4) the coding sequence encodes a peptide more than 6 amino acids long – we call these translated sequences as main coding sequences or mCDS.

## Modeling ribosome footprints of different lengths

We observed that ribosome footprints with different lengths, arising due to incomplete nuclease digestion, show slightly different patterns of abundance when aggregated across transcripts (see *Figure 1—figure supplement 6*). To model these differences, we partitioned the footprints into multiple groups based on length, and modeled the data in each group with a separate set of parameters in the emission probability (all groups share the same state sequence along a transcript). Specifically, for $G$ groups of footprints, the data at the $m^{th}$ triplet in $f^{th}$ reading frame $X_{f,m}^n$ can be partitioned into $G$ components, $X_{f,m}^n = \left(X_{g,fm}^n\right)_{g=1}^G$, where $X_{g,fm}^n$ denotes the triplet of RPF counts from $g^{\text{th}}$ group. Assuming that the RPF counts from different groups at a given triplet are

independent, conditional on the state of the triplet, the emission probability can be written as $\mathrm{P}\left(X_{f,m}^n|Z_{f,m}^n=z\right)=\prod_{g=1}^G \mathrm{P}\left(X_{g,fm}^n|Z_{f,m}^n=z\right)$ and

$$X_{g,fm}^n|Y_{g,fm}^n,Z_{f,m}^n=z \sim \mathrm{multinomial}\left(Y_{g,fm}^n,\pi_{g,z}\right), \tag{13}$$

$$Y_{g,fm}^n|Z_{f,m}^n=z \sim \mathrm{Poisson}\left(\mu_{g,zfm}^n E^n\right), \tag{14}$$

$$\mu_{g,zfm}^n \sim \mathrm{gamma}\left(\alpha_{g,z},\beta_{g,z}\right), \tag{15}$$

where group-specific parameters, $\left(\pi_{g,z},\alpha_{g,z},\beta_{g,z}\right)$, capture the distinct patterns in each group. The RPF data used in our analyses had four groups of footprints of lengths 28, 29, 30, and 31 bases.

## Base positions with missing data

Approximately 15% of the transcriptome have unmappable base positions, in part due to the short lengths of ribosome footprints. Consider the $m^{th}$ base triplet in frame $f$ in the $n^{th}$ transcript. If $J_{g,fm}^n$ is the set of positions in this triplet that are unmappable for footprints corresponding to group $g$, the emission probabilities become

$$X_{g,fm}^n|Y_{g,fm}^n,Z_{f,m}^n=z \sim \mathrm{multinomial}\left(Y_{g,fm}^n,\tilde{\pi}_{g,z}\right), \tag{16}$$

$$Y_{g,fm}^n|Z_{f,m}^n=z \sim \mathrm{Poisson}\left(\psi_{g,zfm}^n\,\mu_{g,zfm}^n E^n\right), \tag{17}$$

$$\mu_{g,zfm}^n \sim \mathrm{gamma}\left(\alpha_{g,z},\beta_{g,z}\right), \tag{18}$$

where

$$\psi_{g,zfm}^n=\sum_{j\notin J_{g,fm}^n}\pi_{g,zj}, \tag{19}$$

$$\tilde{\pi}_{g,zj}=\begin{cases}0 & if\ j\in J_{g,fm}^n\\ \frac{\pi_{g,zj}}{\psi_{g,zfm}^n} & \text{otherwise.}\end{cases} \tag{20}$$

If all three positions in a triplet are unmappable, then we treat the triplet as having missing data for that footprint group and set $\mathrm{P}\left(X_{g,fm}^n|Z_{f,m}^n\right)=1$ for all values of $Z_{f,m}^n$.

## Kozak model

Using the annotated initiation sites of GENCODE annotated coding transcripts, we estimated a position weight matrix (PWM) that captures the base composition of the $-9$ to $+6$ positions flanking known initiation sites. Since the consensus sequence of this PWM is the same as the reported consensus Kozak sequence (*Kozak, 1987*), we refer to this model as the Kozak model. We estimated a background PWM model using the same set of positions relative to random AUG triplets within the same set of transcripts. For the $m^{th}$ triplet in frame $f$ in the $n^{th}$ transcript, using the base sequence from the -9 to +6 positions flanking this triplet, we computed $K_{f,m}^n$, the log of ratio of likelihood of the flanking sequence under the Kozak model to likelihood under the background model.

## Transcripts with length not a multiple of three

The length of such a transcript can be written as $L_n=3M_n+B$, where $B\in\{1,2\}$. We assumed that the RPF data on the first $3M_n$ bases $\left(T_{1:3M_n}^n\right)$ and the data on the remaining $B$ bases $\left(T_{3M_n+1:L_n}^n\right)$ are independent. We modeled $T_{1:3M_n}^n$ using a mixture of HMMs as described above, and modeled $T_{3M_n+1:L_n}^n$ using the emission probability of the $3'\mathrm{UTS}$ state as follows.

$$\mathrm{P}\left(T_{3M_n+1:L_n}^n | E^n, \alpha_z, \beta_z\right) = \prod_{m=3M_n+1}^{L_n} \mathrm{P}\left(T_m^n | E^n, \alpha_z, \beta_z\right), \tag{21}$$

$$T_m^n \sim \mathrm{Poisson}\left(\frac{1}{3}\tau_m^n E^n\right), \tag{22}$$

$$\tau_m^n \sim \mathrm{gamma}\left(\alpha_z, \beta_z\right), \tag{23}$$

$$z = 3'\mathrm{UTS}$$

A Python implementation of riboHMM can be downloaded from https://rajanil.github.io/riboHMM/.

## Quantifying false discoveries of riboHMM

We characterize the performance of riboHMM by addressing three scenarios: (1) How often does riboHMM identify an mCDS in transcripts with no signal of translation? (2) How often does riboHMM identify an incorrect reading frame in transcripts with signal for translation? (3) When riboHMM identifies the correct reading frame in transcripts with signal for translation, how often does it identify an incorrect initiation site? To address the first question, we started with the transcripts for which riboHMM was able to identify an mCDS and generated a set of "null transcripts" by permuting the footprint counts among base positions within each transcript. Applying a posterior cutoff of 0.8, riboHMM incorrectly identified an mCDS in 4.5% of these null transcripts. We used this estimate of the Type 1 error rate to compute the false discovery rate for novel mCDS in noncoding transcripts and novel uaCDS identified by riboHMM. To address the other two questions, we started with the set of annotated coding transcripts for which riboHMM was able to recover the precise CDS (i.e., the mCDS matched the annotated CDS exactly). We generated a set of "simulated transcripts" using the following strategy: (1) randomly select a new TIS downstream and in-frame to the annotated TIS, ensuring that the codon underlying the new TIS belonged to the set $\Omega_{\mathrm{start}}$, (2) permute the footprint counts among bases upstream of the new TIS. Among the simulated transcripts in which riboHMM could identify an mCDS, the inferred reading frame was completely different from the true translated reading frame in 0.31% transcripts. We used this estimate of the Type 1 error rate to quantify false discoveries among novel mCDS in annotated coding transcripts. In the remaining simulated transcripts, the inferred TIS matched the new TIS exactly in 62% of transcripts; this corresponds to a false discovery proportion of 38%.

## Translated mCDS in pseudogenes

Starting with 14,065 pseudogenes that have been identified and categorized in humans (*Pei et al., 2012*), 9,375 pseudogenes were identified by StringTie to be expressed in LCLs. Using a very stringent posterior cutoff of 99.99%, we inferred mCDS in 448 of these expressed pseudogenes. Using pairwise alignment of the pseudogene and parent gene transcript, we observed that although the pseudogene mCDS typically code for shorter protein sequences compared with the parent protein, a large fraction of the pseudogene mCDS share coding-frame with their parent gene (see *Figure 3—figure supplement 1*).

## Validation with Harringtonine-treated data

Harringtonine-treated ribosome footprints were measured in LCLs with a total sequencing depth of 21 million reads. In *Figure 4*, we illustrate the aggregate proportion of treated ribosome footprints centered at the inferred start codon for all novel mCDS, and compare it with the aggregate proportion of treated footprints around the start codon of an equal number of annotated CDSs that have a posterior probability greater than 0.8 under our model. In *Figure 4—figure supplement 1*, we illustrate the aggregate proportion of treated footprints for mCDS inferred in pseudogenes alone, and in *Figure 6B*, we compare the aggregate treated footprint proportions at the start codons of inferred uaCDS and their corresponding mCDS.

## Identifying translated alternate ORFs

For each transcript that had a mCDS with posterior greater than 0.8 and more than 50 base pairs of RNA sequence in the 5'UTS state, we defined an "upstream-restricted transcript" consisting of the exons within the 5'UTS state. Using a random set of 5000 non-overlapping upstream-restricted transcripts in which more than 80% of base positions were mappable, we computed the maximum likelihood estimates of the transition parameters and occupancy parameters to identify additional translated sequences within these upstream-restricted transcripts. Assuming that the fine-scale structure of footprint counts within these translated sequences would be similar to that within the mCDS, we kept the periodicity parameters fixed to their previously estimated values. With these parameter estimates, we inferred the MAP frame and state sequences with posterior greater than 0.8 and filtered out inferences where less than 50% of the inferred CDS was mappable. These additional translated sequences within the upstream-restricted transcripts were called upstream alternate coding sequences or uaCDS.

## Identifying stable peptides with mass spectrometry data

To identify stable proteins translated from the novel CDSs (mCDS and uaCDS), we analyzed quantitative, high-resolution mass spectrometry data derived from 60 LCLs, with MaxQuant v1.5.0.30 (*Cox and Mann, 2008*) and the Andromeda (*Cox et al., 2011*) search engine. Sample labeling, processing and data collection details can be found elsewhere (*Battle et al., 2015*; *Khan et al., 2013*). Peptides were identified using a database that contained 63,904 GENCODE annotated protein sequences and 7271 novel CDSs identified by our method. For all searches, up to two missed tryptic cleavages were allowed, carbamidomethylation of cysteine was entered as a fixed modification, and N-terminal acetylation and oxidation of methionine were included as variable modifications for all searches. A 'first search' tolerance of 40 ppm with a score threshold of 70 was used for time-dependent mass recalibration followed by a main search MS1 tolerance of 6 ppm and an MS2 tolerance of 20 ppm. The 're-quantify' option was used to aid the assignment of isotopic patterns to labeling pairs. The 'match between runs' option was enabled to match identifications across samples using a matching time window of 42 s and an alignment time window of 20 min. Peptide and protein false discovery rates were set to 10% using a reverted version of the search database. Protein group quantifications were taken as the median $\log_2$ (sample/standard) ratio for all groups containing at least two independent unique or 'razor' peptide quantifications (including multiple measurements of the same peptide in different fractions) without a modified peptide counterpart.

## Bias correction to compute expected number of peptide hits

Proteins with at least one peptide identified by this high-resolution mass-spectrometry protocol tend to be distinct from proteins with no mass-spectrum matches.

1. 1. The median footprint density of annotated coding genes with at least one peptide match is about 125 fold higher than that of coding genes with no peptide match (see *Figure 3—figure supplement 4A*).
2. The median length of coding genes with at least one peptide match is 20% higher than that of coding genes without a peptide match (see *Figure 3—figure supplement 4B*).
3. 3. The trypsin cleavage step of the protocol ensures that nearly all observable peptides have a C-terminal lysine or arginine residue, and up to two additional lysine or arginine residues within the peptide sequence (called "tryptic peptides"). This step imposes a strict constraint on the set of unique peptide sequences that can be observed from a protein sequence, and genes with fewer tryptic peptides are less likely to have a mass-spectrum match.
4. All tryptic peptides in an expressed protein are not equally likely to be observed. The probability of detecting a tryptic peptide depends on its electrostatic properties relative to other tryptic peptides from all expressed proteins, which in turn depends on the amino acid composition of the tryptic peptides (see *Figure 3—figure supplement 4C–F*).

To account for these biases, we developed a predictive model to estimate the probability that a protein has at least one peptide hit in a mass-spectrometry experiment. The predicted label for a protein is whether the protein has at least one mass-spectrum match ($H_n = 1$) or no mass-spectrum match ($H_n = 0$). The predictive features of a protein used in the model are (1) the ribosome footprint density of the corresponding transcript ($D_n$), (2) the protein length ($S_n$), and (3) the counts of amino

acids within each of the $K$ tryptic peptides that can be generated from the protein ($L_n = \{L_{n1}, \cdots, L_{nK}\}$). Since the relevant feature of an amino acid is its charge, we partitioned the set of amino acids into four groups – positively charged (R, H, K), negatively charged (D, E), polar uncharged (S, T, N Q), and others. The amino acid count vector $L_{nk}$ was then collapsed into a vector of the counts of each of these four groups. Conditional on $H_n = 1$, we introduced a latent variable for each tryptic peptide that indicates whether the peptide was matched to a mass-spectrum or not ($Z_{nk} \in \{1, 0\}$); this latent variable accounts for differences between matched and unmatched peptides.

Assuming that the three predictive features are independent conditional on the predicted label $H_n$, the odds of observing at least one peptide hit is then given as

$$\frac{p(H_n = 1 | D_n, S_n, L_n)}{p(H_n = 0 | D_n, S_n, L_n)} = \frac{p(D_n | H_n = 1)\, p(S_n | H_n = 1) \left\{ \sum_{Z_n} p(L_n | Z_n, H_n = 1) p(Z_n | H_n = 1) \right\} p(H_n = 1)}{p(D_n | H_n = 0)\, p(S_n | H_n = 0)\, p(L_n | Z_n = 0, H_n = 0)\, p(H_n = 0)}$$

We learn the predictive model using annotated coding genes and partitioning them into those that have at least one peptide hit ("hit genes") and those that do not have a peptide hit ("no-hit genes"). We computed $p(D_n | H_n)$ using an empirical distribution of footprint density within coding genes, $p(S_n | H_n)$ using an empirical distribution of the lengths of coding genes, $p(L_{nk} | Z_{nk} = 1, H_n = 1)$ using tryptic peptides within hit genes matched to mass-spectra, $p(L_{nk} | Z_{nk} = 0, H_n = 1)$ using unmatched tryptic peptides within hit genes, and $p(L_{nk} | Z_{nk} = 0, H_n = 0)$ using tryptic peptides within no-hit genes. Finally, we set $p(H_n = 1) = p(H_n = 0) = 1/2$ and $p(Z_n | H_n = 1) = 1/(2^K - 1)$. Using peptide hits in annotated proteins, we evaluated the accuracy of this model by holding out some annotated proteins as test data, learning the predictive distributions using the remaining training data and computing the expected number of test proteins that had a mass-spectrum match. We estimated the expected number of held-out annotated proteins with at least one mass-spectrum match to be 1206 (s.d. = 34), while the actual number of held-out proteins with a match was 1387 (s.d. = 36).

## Test for long-term purifying selection

In order to quantify whether the novel mCDS are evolutionary conserved in terms of their amino acid sequence, we first extracted DNA sequences orthologous to the mCDS from a 100-way vertebrate whole-genome alignments (UCSC), restricting to genomes aligned with either Syntenic net or Reciprocal best net. We next performed a three-frame translation on each orthologous sequence and a multiple alignment to obtain the correct codon alignments. More specifically, for each orthologous sequence, we kept the frame with the highest amino acid identity compared to the human peptide, requiring at least 60% identity for alignable positions and no more than 50% of the alignment as gaps. Finally, we used codeML/PAML (*Yang, 2007*) to estimate *dN* and *dS* rates across the trees consisting of all remaining peptides, first using a model allowing variable omega and then a model with omega fixed to one. To determine whether a specific peptide is under purifying selection or not, we compared the two models using a likelihood ratio test and reported peptides that satisfied a Bonferroni-corrected *p*-value threshold.

## Correlation between uaCDS and mCDS

We computed the correlation across LCLs between the proportion of footprints mapped to a transcript that fall within its uaCDS and the proportion that fall within its mCDS. We evaluated the statistical significance of these correlations using an empirical null distribution of Spearman correlations computed using random pairs of mCDS and uaCDS. A random pair of mCDS and uaCDS was obtained by randomly shifting the coordinates of an observed pair of mCDS and uaCDS, matching for their respective lengths and the distance between them.

### Data release

All novel coding sequences identified in this work, along with the harringtonine-treated ribosome profiling data are deposited in GEO Accession GSE75290.

## Acknowledgements

We thank Adam Frankish from the GENCODE group for discussions on the sources of information used for annotating coding genes, Zia Khan for discussions on the mass-spectrometry data analysis, and Audrey Fu for discussions on evaluating the correlation between mCDS and uaCDS. We also thank members of the Pritchard, Gilad and Stephens labs for comments and suggestions, and the two anonymous reviewers, the Reviewing Editor, and Naama Barkai, the Senior Editor for their insightful reviews and comments. This work was funded by grants from the NIH (HG007036 to JKP, MH084703 to YG and JKP, and HG02585 to MS), and by the Howard Hughes Medical Institute. The funders had no role in study design, data collection and analysis, decision to publish, or preparation of the manuscript.

## Additional information

### Funding

| Funder | Grant reference number | Author |
| --- | --- | --- |
| National Institutes of Health | HG007036 | Jonathan K Pritchard |
| National Institutes of Health | MH084703 | Yoav Gilad<br>Jonathan K Pritchard |
| National Institutes of Health | HG02585 | Matthew Stephens |
| Howard Hughes Medical Institute | | Jonathan K Pritchard |

The funders had no role in study design, data collection and interpretation, or the decision to submit the work for publication.

### Author contributions

AR, HS, MS, YG, JKP, Conception and design, Analysis and interpretation of data, Drafting or revising the article; SHW, Conception and design, Acquisition of data, Analysis and interpretation of data, Drafting or revising the article; AH, YIL, BE, Analysis and interpretation of data, Drafting or revising the article

### Author ORCIDs

Anil Raj, http://orcid.org/0000-0003-4412-0883
Yang I Li, http://orcid.org/0000-0002-0736-251X
Brett Engelmann, http://orcid.org/0000-0002-9845-6668

## Additional files

### Supplementary files

• Supplementary file 1. Peptide matches to novel CDS detected using mass spectrometry. This table lists the 207 novel CDS (161 mCDS and 46 uaCDS) that have at least one mass-spectrum uniquely matching a peptide in the inferred protein sequence, at protein-level 10% FDR.

### Major datasets

The following dataset was generated:

| Author(s) | Year | Dataset title | Dataset URL | Database, license, and accessibility information |
| --- | --- | --- | --- | --- |
| Wang SH, Raj A, Shim H, Gilad Y, Pritchard JK, Stephens M, Engelmann B, Li YI, Harpak A | 2015 | Thousands of novel translated open reading frames in humans inferred by ribosome footprint profiling | http://www.ncbi.nlm.nih.gov/geo/query/acc.cgi?acc=GSE75290 | Publicly available at the NCBI Gene Expression Omnibus (accession no: GSE75290). |

The following previously published datasets were used:

| Author(s) | Year | Dataset title | Dataset URL | Database, license, and accessibility information |
|---|---|---|---|---|
| Khan Z | 2015 | Mass-spectrometry measurements in 60 human LCLs | http://proteomecentral.proteomexchange.org/cgi/GetDataset?ID=PXD001406 | Publicly available at ProteomeXchange (accession no: PXD001406) |
| Battle A, Khan Z, Wang SH, Mitrano A, Ford MJ, Pritchard JK, Gilad Y | 2015 | Impact of regulatory variation from RNA to protein | http://www.ncbi.nlm.nih.gov/geo/query/acc.cgi?acc=GSE61742 | Publicly available at the NCBI Gene Expression Omnibus (accession no: GSE61742) |
| Lappalainen T, Dermitzakis E | 2013 | RNA-seq measurementsin 86 human LCLs | http://www.ebi.ac.uk/arrayexpress/experiments/E-GEUV-1/ | Publicly available at EMBL European Bioinformatics Institute (accession no: E-GEUV-1) |

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
