## [Decision Letter]

Thank you for submitting your work entitled "Thousands of novel translated open reading frames in humans inferred by ribosome footprint profiling" for consideration by *eLife*. Your article has been favorably evaluated by Naama Barkai (Senior editor) and two reviewers, one of whom served as guest Reviewing Editor.

The reviewers have discussed the reviews with one another and the Reviewing Editor has drafted this decision to help you prepare a revised submission.

Summary:

Ribosome profiling seems to promise data that will allow empirical and unbiased detection of translated reading frames. In this manuscript, the authors apply a novel algorithm based on hidden markov models to analyze ribosome profiling data for a panel of human lymphoblastoid cell lines which they generated as part of this study. The new dataset is interesting, the computational approach is sensible and seems sound.

The end product of the analysis is a collection over 7,000 coding sequences (CDS) that are predicted to be translated to protein by the cell. They identify a wide variety of novel coding sequences, reflecting alternate reading frames on known mRNAs as well as translation of unannotated transcripts and pseudogenes. The authors try to demonstrate the validity of these predictions in various ways. The first comparison uses data generated in the presence of a drug that arrests the ribosome, which allows that start position and reading frame of the predicted CDS to be assessed. The p-values reported for the statistical significance in aggregate are impressive, but the underlying data in Figure 4 suggest that the false positive rate is high. The second approach to validation considers the rate of non-synonymous vs synonymous substitutions inside the predicted CDS. Again, only for a small fraction (roughly 5%) of the CDS predictions there is statistical evidence that evolutionary pressure keeps the corresponding protein sequence constant, and the size of the effect for novel CDS in Figure 5 is tiny. A third comparison, with mass spec data for protein sequences, again reveals a confirmation rate (5%) that falls far short of what would be expected for matching annotated CDS.

Conceptually, the present work offers a statistically well-grounded approach for identifying translated reading frames. The results include a few interesting biological insights: alternative non-AUG start codons occur much more often than expected, a significant fraction of CDS have upstream alternatives whose level is mutually anti-correlated, and cis-QTLs can be mapped for the relative preference between the two alternatives. Empirically, the authors are able to ground their novel peptide predictions in direct detection of translated protein products, albeit at a rather low detection efficiency. These advances distinguish the present work from other recent studies addressing the same question, and on these grounds merit publication in *eLife*.

This is an excellent study, and the authors have done the best they can to analyze the data. It is perhaps disappointing that such a small fraction of the novel CDS predictions seem to hold up in validation. The authors speculate that these peptides may turn over more rapidly than for annotated CDS (perhaps by analogy with antisense transcription). This is indeed a possible explanation, but it would require further substantiation, which is beyond the scope of this study.

Essential revisions:

1) There is a risk that the larger community will use these new annotations as a resource without being aware of the low validation rate at the level of individual CDS. I therefore feel strongly that the authors should address the ~5% validation rate explicitly in the Abstract. That said, the validation rate in terms of transcriptional initiation may be much more favorable than that in terms of steady-state protein abundance. Future studies will need to address this further, but this is still an important work that sets a significant step towards a more comprehensive understanding of translational control on a genome-wide scale.

2) The authors estimate model parameters, including start codon usage psi_c, from five thousand well-expressed, annotated CDSes. However, annotated mCDSes, cryptic/novel mCDSes, and uaCDSes may differ – and in particular annotated mCDSes likely show a particularly strong bias towards the use of AUG codons, and a near-complete absence of most others besides CUG, which occurs in a few specific genes such as c-Myc. Does this effect in the estimated psi_c values bias the discovery of new reading frames in order to produce the trend shown in Figure 3?

3) The authors report that 310 / 7,801 GENCODE-analyzed genes showed an "entirely distinct" mCDS. However, they estimate their per-transcript Type I error rate as 4.5%, which seems roughly consistent with all 310 instances of distinct CDSes reflecting annotation errors.

4) Identification of the precise translation initiation site has a higher false discovery rate than overall CDS identification. Other recent work (e.g. Fields et al.) incorporates harringtonine start site profiling directly into CDS predictions. Could the authors take a similar approach to improve detection of the correct initiation site (and perhaps thereby improve overall CDS accuracy too)?

5) As a related point, is there a general trend to identify CDSes that are longer, or CDSes that are shorter, when identifying the correct reading frame but not the annotated start site?

6) In the second paragraph of the subsection “Translation of short alternate coding sequences in addition to the mCDS”, the authors report 46 uaCDSes with detected peptides and 317 uaCDSes with evidence for selective constraint – are these two groups correlated?

---

## [Author Response]

*Essential revisions:*

1) There is a risk that the larger community will use these new annotations as a resource without being aware of the low validation rate at the level of individual CDS. I therefore feel strongly that the authors should address the ~5% validation rate explicitly in the Abstract. That said, the validation rate in terms of transcriptional initiation may be much more favorable than that in terms of steady-state protein abundance. Future studies will need to address this further, but this is still an important work that sets a significant step towards a more comprehensive understanding of translational control on a genome-wide scale.

We have updated the Abstract to highlight the validation rate using mass-spectrometry measurements. That said, we note that the class of novel coding sequences is drastically different from annotated coding sequences in the lengths of the final protein. Thus, standard approaches to estimating selective constraint are likely to be significantly underpowered for these novel proteins. Moreover, standard mass-spectrometry measurements are not designed to measure proteins of these sizes and translation levels. We think it is important to quantify the observed number of validated novel CDS against a number that we can reasonably expect to be validated (to account for possible biases in the validation methods). Based on this, the mass-spectrometry validation rate for all novel coding sequences is 14%, increasing to 27% for just the class of mCDS. Details on how we established the expected novel CDS validation are described in the Materials and methods section.

2) The authors estimate model parameters, including start codon usage psi_c, from five thousand well-expressed, annotated CDSes. However, annotated mCDSes, cryptic/novel mCDSes, and uaCDSes may differ – and in particular annotated mCDSes likely show a particularly strong bias towards the use of AUG codons, and a near-complete absence of most others besides CUG, which occurs in a few specific genes such as c-Myc. Does this effect in the estimated psi_c values bias the discovery of new reading frames in order to produce the trend shown in Figure 3?

Our model makes the assumption that all cytoplasmic mRNA translation shares the same molecular mechanism. Our learning set is enriched for annotated coding genes and we agree that this choice of learning set could lead to bias in estimating the frequency of non-AUG initiated coding sequences. Given the general lack of available information on these novel CDS, we find our estimate a reasonable starting point. The model parameters were learned using five thousand well-expressed genes; we used GENCODE annotations to define a gene but did not restrict this set to annotated coding genes. Most importantly, we did not use any information about the annotated translation initiation and termination sites when learning the model parameters. The psi_c values estimated with these well-expressed 5000 genes were very similar to those observed when random sets of five thousand genes were used as the learning set (see Figure 8). In addition, only the periodicity parameters are shared between uaCDS and mCDS; i.e., we re-estimate psi_c before inferring uaCDS. We decided to use well-expressed genes as our learning set to ensure that when the footprint data do not provide very strong evidence regarding the initiation site, novel coding sequences identified by our method are as similar as possible to annotated coding sequences in the sequence composition of their initiation sites. We have now elaborated on our choice of learning set and its consequences in the Discussion section of the revised manuscript (third paragraph).

Author response image 1.Comparing the estimated values of parameter psi_c, when using the 5000 most expressed genes (blue) and random sets of 5000 genes (gray).**DOI:**
http://dx.doi.org/10.7554/eLife.13328.025

3) The authors report that 310 / 7,801 GENCODE-analyzed genes showed an "entirely distinct" mCDS. However, they estimate their per-transcript Type I error rate as 4.5%, which seems roughly consistent with all 310 instances of distinct CDSes reflecting annotation errors.

For annotated coding transcripts, the relevant Type I error rate for riboHMM is defined as follows: given that a set of transcripts each have a specific translated reading frame, the Type I error rate is the fraction of transcripts in which riboHMM infers a completely distinct reading frame as translated. Using simulations (described in detail in the subsection “Quantifying false discoveries of riboHMM”), we estimated the Type I error rate for annotated coding transcripts to be 0.31%. Thus, given that annotated coding transcripts have a specific translated reading frame, our error rate in inferring a completely different reading frame as translated is very small. The Type I error rate of 4.5% is more relevant to annotated non-coding transcripts. We have updated the manuscript clarifying the Type I error rate for these instances in the last paragraph of the Introduction and describing it in the subsection “Quantifying false discoveries of riboHMM” in Materials and methods.

As additional evidence supporting our inferences for these instances, Figure 9 shows that the aggregate harringtonine footprint signal at the annotated initiation sites is substantially lower than the signal at the initiation sites inferred by our model, for the same set of annotated coding transcripts.

Author response image 2.Comparing the proportion of harringtonine-treated ribosome footprints at the inferred initiation sites of novel coding sequences and the annotated initiation sites of annotated coding sequences.Both plots are an aggregate over the same set of annotated coding genes for which the inferred coding sequence does not match the annotated coding sequence.**DOI:**
http://dx.doi.org/10.7554/eLife.13328.026

4) Identification of the precise translation initiation site has a higher false discovery rate than overall CDS identification. Other recent work (e.g. Fields et al.) incorporates harringtonine start site profiling directly into CDS predictions. Could the authors take a similar approach to improve detection of the correct initiation site (and perhaps thereby improve overall CDS accuracy too)?

We think it is a great idea to use the harringtonine footprint signal around the inferred initiation sites to improve overall accuracy of inferring novel coding sequences, similar to the approach in Fields et al. We attempted to incorporate harringtonine treated data in the model by introducing an additional covariate in the transition probabilities, providing independent information on the positions of translation initiation sites. However, the codon usage at the inferred initiation sites showed no significant change (K-S test; p-value = 0.88) and the set of inferred coding sequences showed very little difference when harringtonine data were incorporated into the model. Since the footprint data without treatment show clear enrichment at initiation sites, it is likely that harringtonine treated data do not provide much additional information. Thus, while the harringtonine data were useful for validating our inferred initiation sites (among potential initiation sites in a transcript) in aggregate, the data did not have sufficient information to calibrate the confidence in our predicted initiation sites across transcripts. We have modified the manuscript to discuss this point in the fourth paragraph of the Discussion section.

5) As a related point, is there a general trend to identify CDSes that are longer, or CDSes that are shorter, when identifying the correct reading frame but not the annotated start site?

When riboHMM identified the correct reading frame but not the annotated start site, we observed that the inferred initiation site was equally likely to be upstream or downstream of the annotated initiation site (Mann-Whitney test; p-value = 0.41). Moreover, riboHMM was no more likely to correctly identify the annotated initiation site in shorter proteins than longer proteins (Mann-Whitney test; p-value = 0.12). We have now included these observations in the second paragraph of the subsection “Detection of novel CDSs in LCLs”.

6) In the second paragraph of the subsection “Translation of short alternate coding sequences in addition to the mCDS”, the authors report 46 uaCDSes with detected peptides and 317 uaCDSes with evidence for selective constraint – are these two groups correlated?

Among uaCDS, those having a peptide match and those having evidence of selective constraint are not concordant (Fisher’s test; p-value = 0.56). This is consistent with the low sensitivity of standard mass-spectrometry protocols to identify very short proteins. We have now noted this at the end of the subsection “Translation of short alternate coding sequences in addition to the mCDS”.